# Global pathogenomic analysis identifies known and candidate genetic antimicrobial resistance determinants in twelve species

Jason C. Hyun[1], Jonathan M. Monk [2], Richard Szubin[2], Ying Hefner[2] & Bernhard O. Palsson [1,2,3,4,5]

Surveillance programs for managing antimicrobial resistance (AMR) have yielded thousands of genomes suited for data-driven mechanism discovery. We present a workflow integrating pangenomics, gene annotation, and machine learning to identify AMR genes at scale. When applied to 12 species, 27,155 genomes, and 69 drugs, we 1) find AMR gene transfer mostly confined within related species, with 925 genes in multiple species but just eight in multiple phylogenetic classes, 2) demonstrate that discovery-oriented support vector machines outperform contemporary methods at recovering known AMR genes, recovering 263 genes compared to 145 by Pyseer, and 3) identify 142 AMR gene candidates. Validation of two candidates in *E. coli* BW25113 reveals cases of conditional resistance: *ΔcycA* confers ciprofloxacin resistance in minimal media with D-serine, and *frdD* V111D confers ampicillin resistance in the presence of *ampC* by modifying the overlapping promoter. We expect this approach to be adaptable to other species and phenotypes.

Antimicrobial resistance (AMR) remains a persistent problem in the treatment of bacterial infections. With resistance having been observed against nearly all major antibiotics[1], 700,000 annual deaths are currently attributable to AMR globally and is projected to increase to as high as 10 million by 2050 without major interventions[2]. One strategy for managing AMR is the large-scale sequencing of infection isolates[3], which has yielded tens of thousands of publicly-available genome sequences for each major bacterial pathogen, frequently paired with resistance metadata[4].

This wealth of data has enabled global analyses on the genetics of AMR, many of which employ machine learning (ML) to predict AMR phenotypes directly from genetic variations[5,6]. Accurate AMR phenotype prediction models trained on thousands of genomes have been developed for many pathogens such as *Escherichia coli*[7,8], *Klebsiella pneumoniae*[9], *Mycobacterium tuberculosis*[10], *Salmonella enterica*[11], or multiple species[12–15], with numerous others developed from smaller datasets. However, many of these studies report a significant fraction

of their models' predictive genetic features to have no relationship to known AMR mechanisms. This disconnect between statistically identified and mechanistically established genetic determinants of AMR highlights the current gap in knowledge in AMR genetics and remains a challenge for the real world adoption of ML-based systems for rapidly predicting AMR and informing treatment strategies[16,17].

Data-centric efforts to close this gap have focused on genome-wide association studies (GWAS), yielding tools for conducting the statistical tests underlying GWAS such as PLINK[18] and GEMMA[19], with some specializing in microbial GWAS by rigorously addressing population structure such as DBGWAS[20] and Pyseer[21]. However, the predictive features identified when training ML models to predict AMR naturally provide another source of AMR gene candidates distinct from GWAS. Such analyses can leverage the extensive body of ML literature and tools to draw from a broader range of statistical and algorithmic frameworks than those previously used for GWAS, and as such, there is a growing effort towards developing ML workflows

[1]Bioinformatics and Systems Biology Program, University of California, San Diego, La Jolla, CA, USA. [2]Department of Bioengineering, University of California, San Diego, La Jolla, CA, USA. [3]Department of Pediatrics, University of California, San Diego, La Jolla, CA, USA. [4]Center for Microbiome Innovation, University of California, San Diego, La Jolla, CA, USA. [5]Novo Nordisk Foundation Center for Biosustainability, Technical University of Denmark, Kemitorvet, Building 220, 2800 Kongens, Lyngby, Denmark. ✉e-mail: palsson@ucsd.edu

specifically aimed at mechanism discovery within AMR genetics[17,22] and beyond[23]. Recent ML-aided GWAS have identified genetic and metabolic mechanisms behind resistance in *M. tuberculosis*[24], and demonstrated ML's competitiveness compared to typical statistical testing at recovering known AMR genes for multiple pathogens[25]. These successes demonstrate the potential for ML to carry out the role of GWAS for the ever-growing public collection of bacterial genome sequences and AMR data.

We present a machine learning analysis tailored towards AMR gene discovery consisting of three components drawn from previous workflows: 1) pangenome construction by sequence clustering to enumerate biologically-interpretable genetic features[26], 2) systematic annotation of known AMR genes among those features with RGI[27], and 3) training of support vector machine (SVM) ensembles to learn relationships between all genetic features and a given AMR phenotype[25]. This workflow was evaluated for both phenotype prediction accuracy and recovery of known AMR genes among predictive features across 12 pathogenic species spanning 127 species-drug combinations and 27,155 genomes. We find that this approach provides both a comprehensive overview of the distribution of known AMR genes and consistently yields ML models that both accurately predict AMR phenotype and outperform contemporary GWAS methods at recovering known AMR genes. Functional analysis of strongly predictive features yielded a set of 142 AMR gene candidates, of which two were experimentally confirmed to impact resistance in *E. coli*.

## Results

### Assembly of twelve bacterial pathogen pangenomes and antimicrobial resistance data

A total of 27,155 genomes across 12 species were downloaded from the PATRIC database[28] after filtering for assembly quality and availability of AMR data (Supplementary Data 1). Pangenomes were constructed and genetic features were enumerated for each species using a sequence clustering approach[26] with CD-HIT v4.6[29] (see Methods), yielding six unique genetic feature types: 1) genes (protein sequence clusters), 2) alleles (protein sequence variants), 3) 5′ variants (300 bp ORF-flanking upstream variants), 4) 3′ variants (300 bp ORF-flanking downstream variants), 5) noncoding feature clusters, and 6) noncoding feature variants (Fig. 1a, b, Supplementary Table 1).

Experimentally derived susceptible-intermediate-resistant (SIR) phenotypes for the genomes were assembled from directly reported SIRs and SIRs inferred from minimum inhibitory concentrations (MICs). MIC breakpoints for SIR inference were determined from genomes with both SIR and MIC values for each species-drug-standard combination (i.e. CLSI, EUCAST) to yield internally-consistent SIR data (see Methods). From 169,693 MICs, 22,772 SIR inferences across 93 species-drug cases were generated for genomes without SIR data. In total, 176,911 SIR phenotypes were assembled across 69 drugs, with 88.3% phenotypes from directly reported SIRs and 11.7% inferred from MICs (Fig. 1c, d, Supplementary Data 2), comprising the largest internally-consistent AMR dataset known to the authors at the time of publication. The overall diversity of the combined genomic and AMR data was evaluated by analyzing the distribution of MLST subtypes, genome BioProject IDs, and susceptible/resistant genomes by drug (Supplementary Fig. 1). Based on these evaluations, the genomes represented here are diverse with respect to subtype, study of origin, and resistance phenotypes, with cases of potentially lower coverage limited to species with AMR data for a smaller range of drugs (*Neisseria gonorrhoeae*) or fewer genomes available in total (*Campylobacter coli, Campylobacter jejuni*) (Supplementary Analysis).

Known AMR genes were identified in each pangenome through direct annotation of alleles by RGI v5.2.0 with CARD ontology v3.1.3[27] and parsing PATRIC text annotations for drug-associated terms. 7710 AMR genes were identified across all species, spanning 95,491 gene-drug mappings (Fig. 1e, Supplementary Data 3). The fewest number of

AMR genes (71) were identified in *Campylobacter coli* and the most in *Escherichia coli* (1533), with the greatest number of AMR genes identified for major drug classes such as beta-lactams, aminoglycosides, and quinolones.

### Global analysis of known AMR genes reveals potential phylogenetic limitations on cross-species gene transfer

To examine the distribution of known AMR genes, all AMR gene alleles across all species were re-clustered with CD-HIT, yielding 6332 unified AMR genes. Rates at which these genes were plasmid- or chromosomally-encoded were predicted by labeling contigs containing AMR genes as plasmid or chromosomal with PlasFlow[30] (Supplementary Data 3). 925 AMR genes were observed in multiple species, with more broadly distributed genes having a greater tendency to be plasmid-encoded (95% of genes in >4 species were plasmid-encoded in a majority of occurrences) (Fig. 2a). Similarly, out of 68,324 unique AMR alleles, 830 were observed in multiple species and were primarily plasmid-encoded (Supplementary Fig. 2a).

Compared against AMR gene categories, specific functions were significantly enriched among both plasmid-encoded (over chromosomal) and multispecies (over single species) AMR genes (Fig. 2b, Supplementary Fig. 3, Supplementary Data 3). Dihydrofolate reductases/ dihydropteroate synthases and aminoglycoside modifying enzymes were significantly enriched by both measures ($n = 6332$ AMR genes, Fisher's exact test, FWER < 0.05, Bonferroni correction, 36 tests), with $\log_2$ odds ratios (LORs) for plasmid over chromosomal genes of 3.0 and 1.8, and LORs for multispecies over single species of 1.5 and 1.3, respectively. Other categories enriched in plasmid but not multispecies genes include chloramphenicol acetyltransferases, ribosomal protection proteins, rRNA methyltransferases, and beta-lactamases (plasmid LOR > 1.0, multispecies LOR < 1.0) (Supplementary Table 2). Generally, multispecies AMR genes tended to be plasmid-encoded and vice versa, with the exception of *rpoB* variants (Fig. 2b). As *rpoB* is a highly conserved chromosomal bacterial gene[31], this exception may be due to many rarely observed *rpoB* fragments on short contigs being misclassified as plasmid-encoded (Supplementary Data 3).

The 925 multispecies AMR genes and 830 AMR alleles were predominantly shared within species of the same phylogenetic class, especially within Gammaproteobacteria (Fig. 2c, Supplementary Fig. 2b). Only 68 (7.4%) multispecies AMR genes and 38 (4.6%) alleles spanned more than one class. Of these, just 8 genes and 5 alleles were observed in at least 10 genomes in each of at least two different phylogenetic classes (Fig. 2d, Supplementary Fig. 2c). These 8 multi-class genes are functionally varied, including TEM family beta-lactamases (*blaTEMs*), ribosomal protection proteins *tetM*, *tetO*, and *tet(W/N/W)*, 23S rRNA methyltransferase *ermB*, aminoglycoside 3′-phosphotransferase *aph(3′)-IIIa*, and lincosamide nucleotidyl-transferase *lnuG*. The *blaTEMs* were observed exclusively on plasmids, while all other multi-class AMR genes were observed on both plasmid and chromosomal DNA. Given the prevalence of *blaTEMs*, we conducted a case study on the distribution of complete *blaTEM* alleles across all species. One variant, TEM-116, was found in gram-positive strains (11 *Staphylococcus aureus* strains), predicted to be on a plasmid shared with *Salmonella enterica* strains, and found among *S. aureus* strains most strongly resistant to cefoxitin (Supplementary Figs. 4–6, Supplementary Table 3, Supplementary Data 4, Supplementary Analysis).

### A GWAS-oriented machine learning approach for the identification of AMR-associated genes outperforms contemporary statistical methods

To identify AMR-associated genetic features, a ML framework was developed to train models for both accuracy at predicting AMR phenotypes and biological relevance, i.e. ability to assign high feature weights to known AMR genes (Fig. 3a). For a given species-drug case,

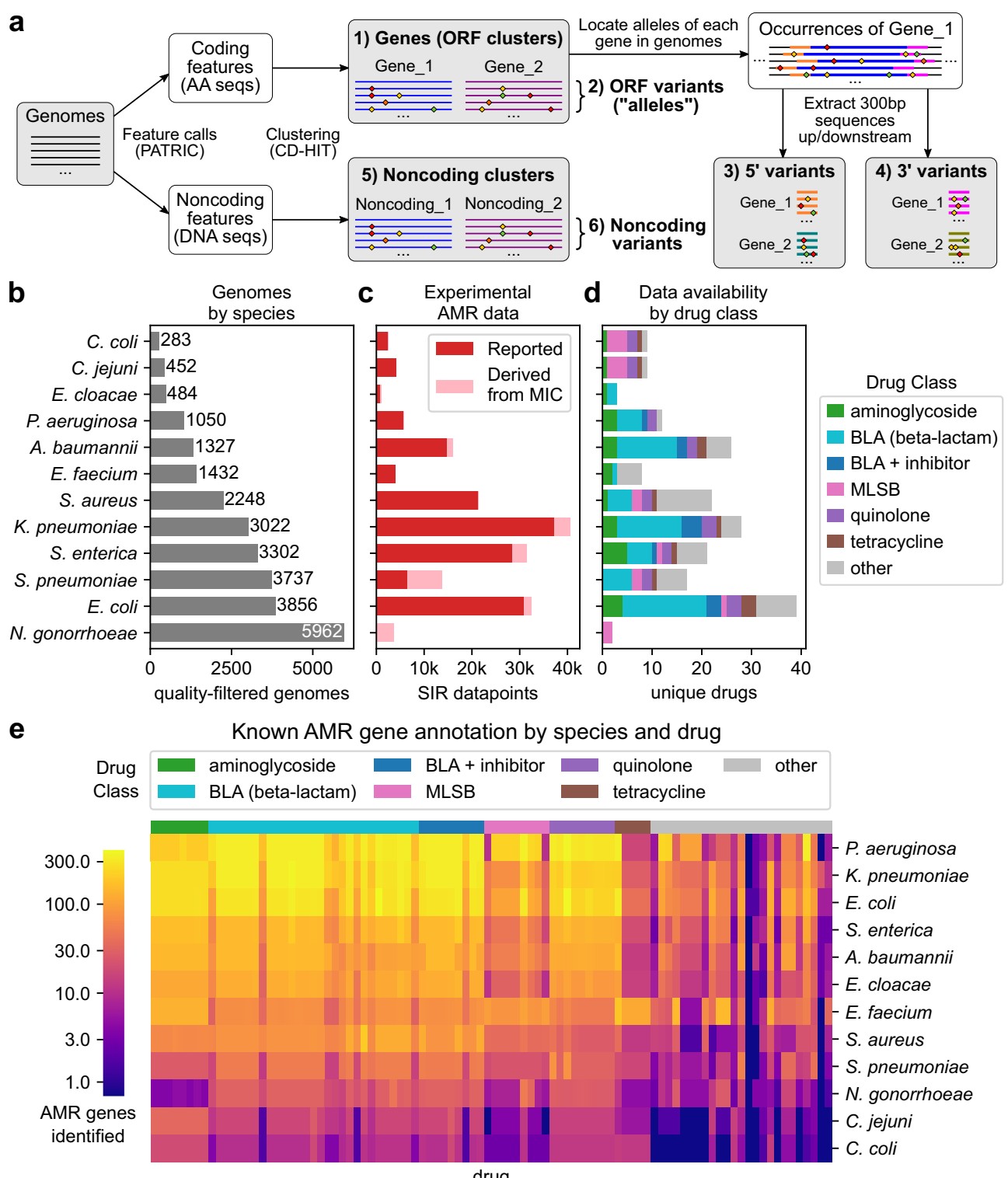

**Fig. 1 | Genomic and antimicrobial resistance datasets assembled from the PATRIC database for 12 pathogenic species. a** Workflow for extracting six types of biological features from a species' genome collection. For each species, the number of (**b**) high-quality genomes with AMR data, (**c**) experimentally measured susceptible-intermediate-resistant (SIR) data points (directly reported or inferred from reported minimum inhibitory concentrations; MICs) with consistent testing standards across all drugs, and (**d**) number of drugs for which SIR data is available is shown. Species have been sorted by number of genomes. The macrolide-lincosamide-streptogramin B drug class is abbreviated MLSB. **e** Number of known AMR genes identified for each species-drug case. Drugs are sorted by drug class, and species are sorted by total number of AMR gene-drug mappings identified.

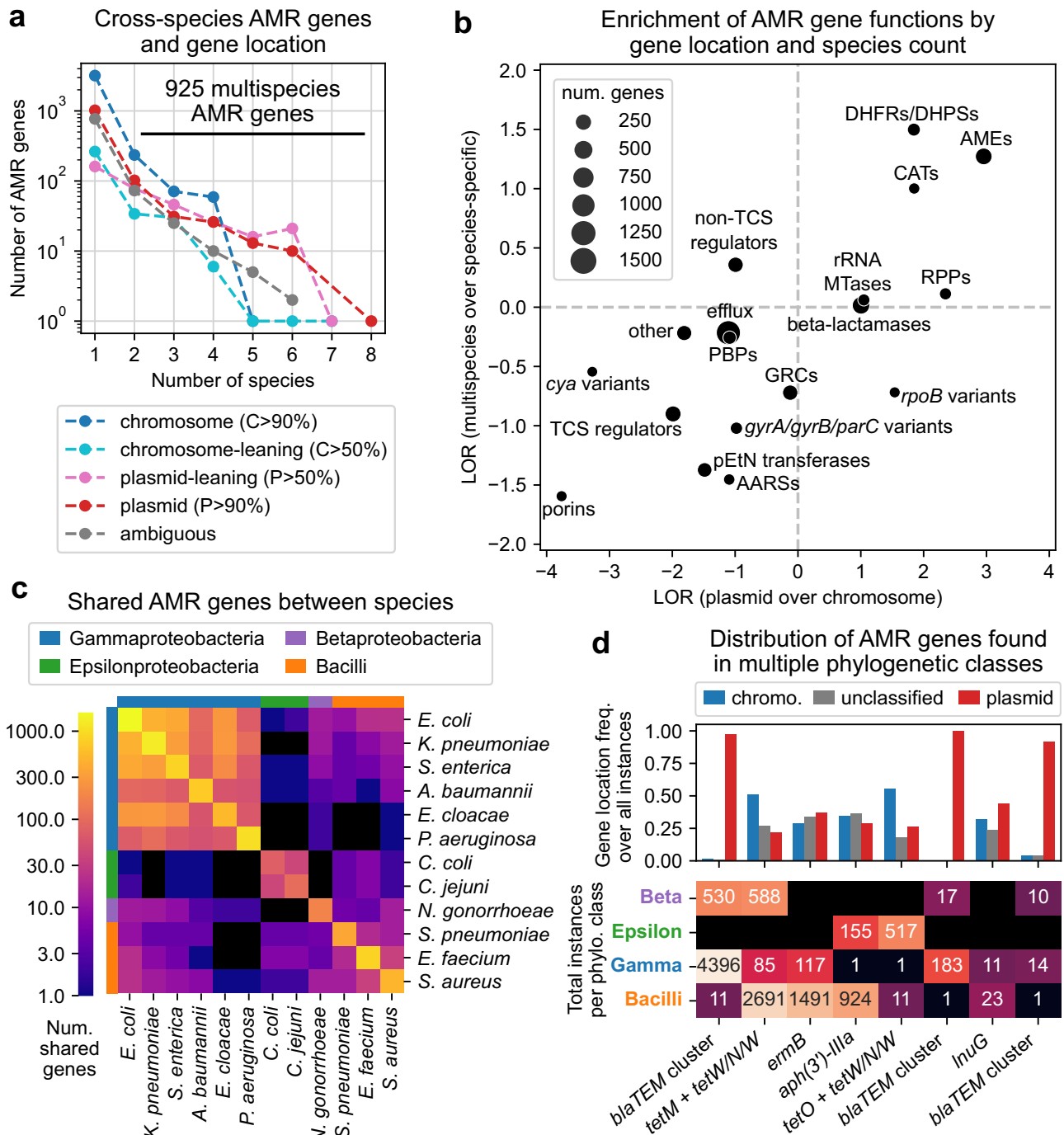

**Fig. 2 | Cross-species analysis of 6332 antimicrobial resistance genes, gene locations, and functions. a** Relationship between the number of species an AMR gene is observed in and tendency to be plasmid-encoded. **b** Enrichment of AMR gene functional categories in plasmid- over chromosomally-encoded genes and in multispecies over single species genes, based on log2 odds ratios (LORs). See Supplementary Data 3 for abbreviations. **c** Number of AMR genes shared between each pair of species, compared to species phylogenetic class. **d** Distribution of predicted gene locations and total occurrences per phylogenetic class for AMR genes appearing in at least 10 genomes of multiple classes. The three *blaTEM* cluster columns correspond to different sequence clusters identified by CD-HIT all related to TEM family beta-lactamases, with the first representing full variants and the others representing fragments.

we started with the support vector machine (SVM) ensemble design described in our previous study[25]. SVM ensembles were trained to classify genomes as susceptible or non-susceptible based on the presence or absence of the genetic features (grouped into six types described previously). Four hyperparameters (HPs), parameters not learned from the data but fixed in advance to control the learning process and model complexity, were varied to evaluate their impact on model performance. Ensembles under various HP combinations were

evaluated over 5-fold cross validation (5CV) for 1) accuracy, as the Matthews correlation coefficient (MCC) on test set genomes to account for class imbalance, and 2) biological relevance, through a "GWAS score" defined as a weighted sum of the rankings of known AMR genetic features after sorting model features by feature weight absolute value. A feature was labeled as a known AMR genetic feature if its corresponding gene cluster was a known AMR gene for the drug of interest as annotated by RGI and PATRIC.

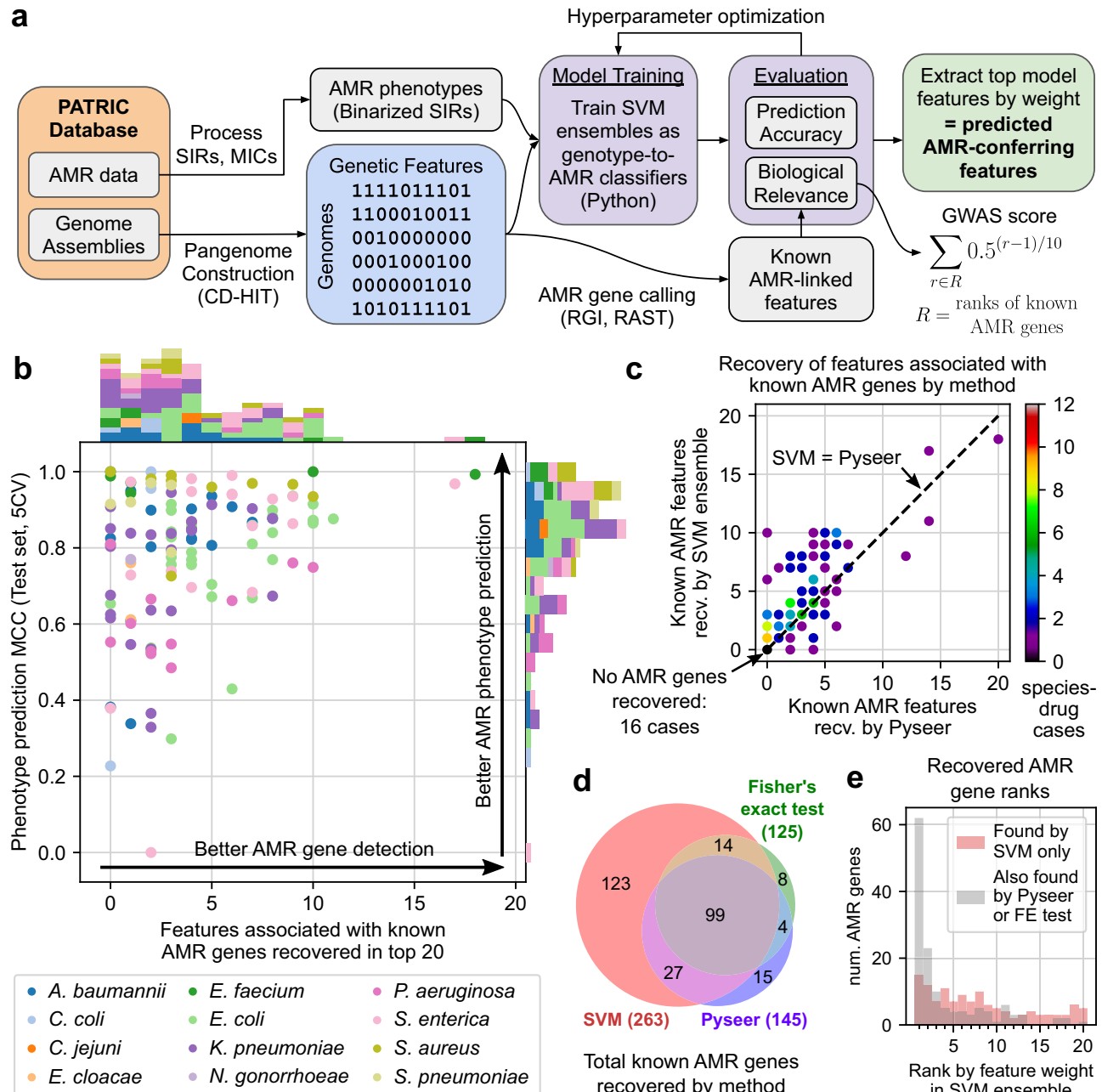

**Fig. 3 | Evaluation of a GWAS-oriented machine learning workflow for identifying AMR-associated genetic features in 127 species-drug cases. a** Workflow integrating support vector machine (SVM) ensembles, hyperparameter optimization, and AMR gene annotation to train models suited for both phenotype prediction and AMR gene identification from genome assemblies and SIR phenotype data. **b** Performance of models for 127 species-drug cases, by mean test set Matthews correlation coefficient (MCC) during 5-fold cross validation and recovery of known AMR genes. **c** Comparison between SVMs and Pyseer at recovering known AMR genes. **d** Total known AMR gene-drug mappings recovered by SVMs, Pyseer, and Fisher's exact tests across all cases. **e** Rankings of known AMR genes recovered among SVMs' top features, grouped by whether or not the gene was also recovered by Pyseer or Fisher's exact test.

HP optimization was first carried out on a set of 10 species-drug test cases, testing 256 HP combinations to eliminate consistently suboptimal HP combinations before scaling to other species-drug cases (Supplementary Figs. 7, 8, Supplementary Tables 4–7, Supplementary Analysis). These results were also used to assess the generalizability of the GWAS score by randomly hiding half of known AMR genes and computing correlations between GWAS scores derived from visible vs. hidden AMR genes. Results across 100 iterations of randomly hiding AMR genes suggests that the GWAS score generalizes well to hidden AMR genes, with Spearman correlation between GWAS scores from visible vs. hidden AMR genes exceeding 0.4 in 6/10

species-drug cases and 0.6 in 4/10 cases (Supplementary Fig. 9, Supplementary Data 5, Supplementary Analysis).

HP optimization was then applied to 127 species-drug cases with at least 100 SIRs, 10 known AMR genes, and minority phenotype >5%. For each case, models under each HP combination were ranked by mean MCC and GWAS score from 5CV, and the HP set with the highest average of the two ranks was selected as optimal. Among the final models, 41 (32%) achieved MCC > 0.9 and 78 (61%) achieved MCC > 0.8 on the test set during 5CV, and 103 models (81%) recovered at least one known AMR genetic feature among the top 20 features (Fig. 3b, Supplementary Data 5). Broadly, a high MCC was necessary but did not

guarantee better recovery of known AMR genetic features, i.e. accuracy did not guarantee biological relevance. Certain dataset parameters were weakly but significantly associated with one or both performance metrics: dataset size, species, fraction of "intermediate" resistant genomes, and number of known AMR genes ($n = 127$, Spearman R or Kruskal–Wallis test, FWER < 0.05, Bonferroni correction, 12 tests, see Supplementary Analysis) (Supplementary Table 8, Supplementary Fig. 10). Finally, relative to models with fixed HPs closest to those in our previous work[25], optimizing HPs offered modest but consistent improvements to both accuracy and known AMR gene recovery. 5CV experiments showed a mean increase in test MCCs and known AMR genes recovered among the top 20 features of 0.035 and 0.230, respectively (Supplementary Fig. 11, Supplementary Data 5). Visualizations of the top 20 model-selected features are available for three example cases (Supplementary Figure 12), and the top 50 features for all 127 models are available in Supplementary Data 5 alongside their associated sequences in Supplementary Data 6.

As a baseline level of AMR gene recovery, for each species-drug case, both Pyseer[21] and Fisher's exact tests were applied to estimate the strength of association between each genetic feature and the AMR phenotype, yielding population-adjusted and unadjusted $p$-values, respectively. Based on the number of known AMR genetic features recovered among the top 20 features (either by feature weight for SVM or $p$-value otherwise), the SVM ensemble approach broadly outperformed both Pyseer and Fisher's exact test. SVM ensembles identified more known AMR-associated features than Pyseer in 73 cases (57%), the same number in 38 cases (30%), and fewer features in just 16 cases (13%) (Fig. 3c, Supplementary Data 5). Nearly half of the equal performance cases (16/38, 42%) were instances where neither method could recover any known AMR features, and similar differences in performance were observed when comparing SVM ensembles and Fisher's exact tests (Supplementary Fig. 13a, Supplementary Data 5). Across all 127 cases, SVM ensembles recovered 263 known AMR gene-drug mappings of which 123 were not recovered by either Pyseer or Fisher's exact tests, compared to just 27 mappings missed by SVM ensemble but recovered by Pyseer or Fisher's exact test (Fig. 3d, Supplementary Data 5). Similar proportions between the number of known AMR gene-drug mappings recovered per method were observed when examining the top 10 or top 50 features (Supplementary Figure 13b, c). Examining SVM ensemble feature rankings, known AMR genes were distributed throughout the full range of ranks among the top 20 features per model, whereas those that were also recovered by other methods were concentrated among the top 3 features (95/138, 69%) with nearly half being the top weighted feature of the corresponding SVM model (62/138, 45%) (Fig. 3e). This result suggests that concordance between these three methods is mostly limited to features with the strongest statistical signals. Finally, a detailed analysis of the 30 $gyrA$ alleles identified by SVM as associated with fluoroquinolone resistance finds all such variants to be consistent with the current literature on $gyrA$-mediated resistance; all positively-associated alleles carried at least one known resistance-conferring substitution, while all negatively-associated alleles carried no such mutations (Supplementary Analysis, Supplementary Data 5).

An examination of the 12 genes recovered by Fisher's exact test but not by SVM ensemble suggests several possible failure modes of the SVM approach (Supplementary Table 9). First, in 4/12 cases, the missed gene is captured slightly outside of the top 20 feature threshold for recovery, with three genes ranked 21 and one ranked 33 by SVM. Second, in another 4/12 cases, many of the top features in the corresponding model were perfectly correlated, saturating the top ranks with these correlates and preventing recovery of additional AMR genes. Third, for 3/12 cases, the corresponding model was not very accurate, with mean test MCC ranging from 0.43 to 0.77. The final remaining case ($arlR$ for ciprofloxacin resistance in $S. aureus$) could not be explained by these previous failure modes.

## Identification of 142 candidate AMR-conferring genes through cross-drug and functional analysis of AMR-predictive features

We next applied several filters to translate the best performing models to a smaller set of high-confidence, AMR-conferring gene candidates (Fig. 4a). Starting with the 78 models achieving test MCC > 0.8, the top 10 features from each model were filtered for those that were not already known AMR genes, occurred in at least 10 genomes with SIR data, and had both positive feature weight and LOR for resistant genomes, yielding 347 features predictive of AMR without known associations to AMR. These candidates were scored based on the number of drugs in the same class for which the feature enriches for resistance and the extent of co-occurrence with known AMR genes. Taking the top 10 features by this score for each species-drug class pair yielded 142 AMR gene candidates (see Methods and Supplementary Data 7).

43 candidates were functionally well-characterized and consisted of four feature types: 16 genes (protein sequence clusters, Table 1), 14 gene alleles (protein sequence variants, Table 2), and 13 gene 5'/3' flanking region variants (Supplementary Table 10). The candidates spanned 8 species and 7 drug classes and majority beta-lactam associated (26/43) due to the relative abundance of beta-lactam AMR data (Fig. 4b). The candidates span many genetic functions, and two functions occurred more than twice. Candidates related to small multidrug resistance (SMR) efflux transporters ($qacE, qacE\Delta1, sugE$) typically associated with resistance to quaternary ammonium compounds[32], were linked to resistance against aminoglycosides, beta-lactams, diaminopyrimidines, and sulfonamides across four species, consistent with previous studies that find SMR transporters associated with resistance against a broad range of antibiotics beyond antiseptics[33,34]. Additionally, three different formate dehydrogenase genes ($fdhF, fdsA, fdnG$) were associated with beta-lactam resistance in $Klebsiella pneumoniae$, suggesting the importance of formate metabolism in AMR, possibly with respect to stress response[35]. Finally, a majority of the sequence-variant level candidates (16/27), especially those related to flanking regions (10/13), were the most common variant of their respective gene clusters, suggesting that most observed perturbations to these genes may be deleterious with respect to AMR.

We selected two allele candidates related to $E. coli$ core genes for experimental validation: The wildtype D-serine/D-alanine/glycine transporter ($cycA$) allele associated with quinolone resistance, and fumarate reductase subunit D ($frdD$) allele with a V111D substitution associated with beta-lactam resistance (see Supplementary Analysis for details on candidate selection). $E. coli$ BW25113 was chosen as the base strain for validation to make use of the Keio knockout collection[36], which is genetically identical to K12 MG1655 across all positions within 40 kb of $frdD$ and $cycA$ based on reference genomes NZ_CP009273.1 and U00096.3. Wildtype (WT) $cycA$ and $frdD$ were defined as the most common allele of the respective genes observed across all $E. coli$ genomes in this study, which were also the alleles present in the BW25113 and K-12 MG1655 reference genomes.

## Experimental validation 1: Loss of amino acid transporter CycA confers limited quinolone resistance in minimal media with D-serine

The WT $cycA$ allele, the fifth highest weighted feature in the $E. coli$ AMR model for levofloxacin, had the highest LOR for resistance among all $cycA$ alleles in 2/4 quinolone drugs (Fig. 5a). To assess the impact of $cycA$ on quinolone resistance, maximum cell density was measured for BW25113 (WT) and corresponding $\Delta cycA$ mutant (KO, from the Keio collection[36]) under 60 conditions based on three variables: concentration of ciprofloxacin (CIP), supplementation with known substrates of the D-serine/D-alanine/glycine transporter encoded by $cycA$[37], and choice of rich vs. minimal media (cation-adjusted Mueller–Hinton Broth CA-MHB vs. M9 media with glucose) (Supplementary Data 8). 6/60 tested conditions resulted in significantly

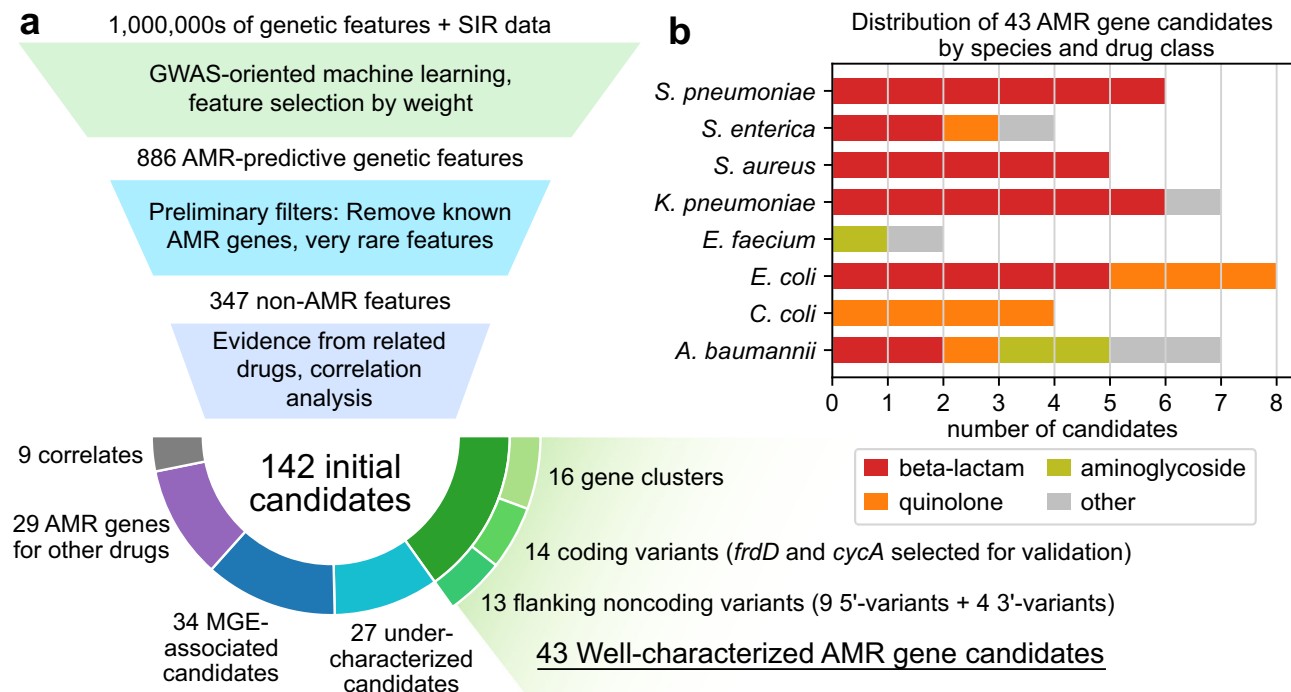

**Fig. 4 | Identification of AMR gene candidates from machine learning models. a** Candidate identification workflow. From each trained AMR-ML model, the top 10 predictive features were identified, filtered for features that are neither known AMR genes nor very rare, ranked based on statistical evidence for resistance in other related drugs, and finally categorized by functional annotation. When multiple features related to the same gene were predictive of resistance, the feature with the strongest evidence was selected and the others were labeled as correlates. Mobile genetic element is abbreviated MGE. **b** Distribution of 43 well-characterized AMR gene candidates by species and drug class.

different final densities between WT and KO ($n_{WT}$=$n_{KO}$ = 3, Welch $t$-test, FDR < 0.05, Benjamini–Hochberg correction, 60 tests), three of which involved D-serine and M9 media (Fig. 5b). In conditions involving D-serine, while increasing CIP concentration reduced final density for both strains, the KO strain was conditionally less sensitive. The KO strain achieved higher densities than WT for 16–125 µg/L CIP, but only in M9 media and not in CA-MHB (Fig. 5c).

One explanation consistent with this conditional increase in CIP resistance by *cycA* KO involves the toxicity of D-serine, its transport by CycA, and interaction with quinolones through the SOS response (Fig. 5d). D-serine, which inhibits L-serine and pantothenate biosynthesis, can be bacteriostatic in minimal media[38]. D-serine uptake can be impaired by *cycA* KO but also through competitive inhibition of CycA by other substrates in rich media[37], and its toxicity mitigated by direct uptake of L-serine and pantothenate in rich media. With respect to CIP, both D-serine[39] and fluoroquinolones[40] induce SOS response but to differing extents. As systematic alterations in SOS response induction have been shown to reduce fluoroquinolone resistance[41], the presence of both D-serine and CIP may result in a SOS response that is neither optimized for D-serine nor CIP, and consequently results in greater susceptibility to CIP.

**Experimental validation 2: The V111D substitution in FrdD confers beta-lactam resistance solely through altering expression of the overlapping beta-lactamase gene *ampC***

Next, the *frdD* V111D allele was selected for validation as it was the third highest weighted feature in the *E. coli* AMR model for ampicillin, in the top 10 for four *E. coli* beta-lactam models, and enriched for resistant strains (LOR > 3) in 7/14 beta-lactam drugs with AMR data (Fig. 6a). Given the proximity of *frdD* to the adjacent beta-lactamase gene *ampC*, this mutation occurs in the −35 box of the *ampC* promoter[42] and coincides with *ampC* overexpression mutations known to increase beta-lactam resistance[43,44] (Fig. 6b). This proximity was confirmed

globally, with 3,698 (96%) *E. coli* genomes harboring both *frdD* and *ampC* on the same contig with ORFs spaced by exactly 63 bp (see Supplementary Analysis). Appropriately, an *ampC* 5′ variant containing the equivalent mutation was ranked 5th in the model for amoxicillin-clavulanate, though no other 5′ variants with the mutation were in the top 50 hits of any other beta-lactam model. To assess whether *frdD* V111D is simply a byproduct of an *ampC* promoter mutation or contributes separately to beta-lactam resistance, we examined two *frdD* mutations both resulting in the V111D substitution but with different effects on *ampC* transcription as predicted by Promoter Calculator[45]: 1) 332 T > A, predicted to increase *ampC* transcription 2.6-fold, or 2) 332TC > AT, predicted to have minimal effect on *ampC* (Fig. 6b). Six *E. coli* strains were examined, based on *frdD* variants (WT or either mutation) generated in either BW25113 or corresponding Δ*ampC* mutant (from the Keio collection[36]). Maximum cell density achieved by these strains were measured under increasing concentrations of ampicillin in either rich (CA-MHB) or minimal (M9 + glucose) media (Supplementary Data 8).

Across all strains, only the mutant with both *ampC* and the overexpression mutation was able to grow at ≥2 mg/L ampicillin in either media, and final densities were not impacted even at 8 mg/L (Fig. 6c). The mutant with *ampC* and the non-overexpressing *frdD* mutation grew at 2 mg/L in CA-MHB only, but this growth was delayed by at least 8 h in all three replicates, and may have resulted from acquiring the single point mutation necessary to yield the known *ampC* overexpressing promoter. Furthermore, all Δ*ampC* mutants failed to grow at ≥2 mg/L ampicillin regardless of *frdD* status and reached lower densities than their corresponding *ampC* WT strain in 17/18 conditions with <2 mg/L ampicillin and significantly so in 15/18 cases ($n_{WT}$=$n_{KO}$ = 3, FDR < 0.05, Welch $t$-test, Benjamini–Hochberg correction, 18 tests, Supplementary Data 8); the only exception was between strains with WT *frdD* at 1 mg/L ampicillin in M9 in which both the *ampC* and *ampC* KO strains exhibited very little growth (OD < 0.1). These results suggest

**Table 1 | 16 gene clusters predicted to be associated with resistance against specific drug classes for individual species**

| Species | Drug class | Accession ID (Gene) | Predicted gene product | Resistant/Susceptible | LORs |
|---|---|---|---|---|---|
| Acinetobacter baumannii | AMG | ENU75377.1 (ydcZ) | Putative inner membrane exporter | GEN = 147/0 AMK = 144/3 TOB = 114/18 | GEN = 7.7 AMK = 5.8 TOB = 1.9 |
| Acinetobacter baumannii | AMG | WP_000679427.1 (qacEΔ1) | Small multidrug resistance (SMR) efflux transporter | GEN = 453/6 AMK = 296/170 TOB = 269/163 | GEN = 4.4 AMK = 1.5 TOB = −0.6 |
| Acinetobacter baumannii | beta-lactam | ACC56111.1 (yfhL) | Uncharacterized ferredoxin-like protein | AMP = 82/0 DOR = 13/0 CFZ = 45/0 | AMP = 6.5 DOR = 6.1 CFZ = 5.6 |
| Enterococcus faecium | AMG | WP_000228166.1 | Nucleotidyltransferase domain containing protein | STR = 89/0 | STR = 14.8 |
| Enterococcus faecium | glycopeptide | WP_000754864.1 (cadC) | Cadmium resistance transcriptional regulatory protein | TEC = 138/0 VAN = 498/150 | TEC = 14.8 VAN = 3.4 |
| Escherichia coli | beta-lactam | WP_000243817.1 | Tryptophan synthase (indole-salvaging) | CXM = 196/1 CTX = 236/4 SAM = 42/0 | CXM = 9.1 CTX = 7.7 SAM = 7.3 |
| Escherichia coli | quinolone | WP_000598813.1 (dcuC) | Anaerobic C4-dicarboxylate transporter | CIP = 59/14 LVX = 31/3 | CIP = 3.3 LVX = 2.1 |
| Klebsiella pneumoniae | beta-lactam | WP_002885150.1 (fdhF) | Formate dehydrogenase H | CRO = 1508/11 ETP = 130/1 CFZ = 1487/58 | CRO = 5.3 ETP = 4.0 CFZ = 3.7 |
| Klebsiella pneumoniae | beta-lactam | AKE78078.1 (fdsA) | Formate dehydrogenase H | CRO = 1499/10 CFZ = 1486/40 ETP = 130/1 | CRO = 5.4 CFZ = 4.4 ETP = 4.0 |
| Klebsiella pneumoniae | beta-lactam | EWF54733.1 (fdnG) | Formate dehydrogenase N alpha subunit | CRO = 1494/10 ETP = 128/1 CFZ = 1478/49 | CRO = 5.4 ETP = 4.0 CFZ = 3.9 |
| Klebsiella pneumoniae | beta-lactam | AHG50656.1 (ygbK) | 3-oxo-tetronate kinase | CRO = 71/0 AMP = 62/0 CAZ = 107/1 | CRO = 6.3 AMP = 6.0 CAZ = 4.0 |
| Klebsiella pneumoniae | diamino-pyrimidine | WP_000679427.1 (qacEΔ1) | Small multidrug resistance (SMR) efflux transporter | TMP = 10/0 SXT = 855/42 | TMP = 6.2 SXT = 3.6 |
| Salmonella enterica | sulfonamide | WP_000800531.1 (qacE) | Small multidrug resistance (SMR) efflux transporter | SXT = 11/4 SIX = 15/0 SMZ = 12/4 | SXT = 6.6 SIX = 5.4 SMZ = 0.4 |
| Staphylococcus aureus | beta-lactam | WP_000872606.1 (maoC) | MaoC domain protein | MET = 202/4 FOX = 786/0 OXA = 29/3 | MET = 16.9 FOX = 16.2 OXA = 15.3 |
| Staphylococcus aureus | beta-lactam | WP_000616816.1 (cadD) | Cadmium resistance transporter | PEN = 461/28 BPG = 82/4 FOX = 330/165 | PEN = 2.0 BPG = 0.1 FOX = −1.5 |
| Staphylococcus aureus | beta-lactam | WP_001186608.1 | Site-specific recombinase | FOX = 661/0 MET = 198/58 OXA = 8/13 | FOX = 12.3 MET = 8.9 OXA = 3.3 |

Accession IDs are provided for the most common sequence variant of each gene cluster (RefSeq when possible, GenBank otherwise), along with gene names when available and gene products. The number of resistant/susceptible genomes and log$_2$ odds ratios (LORs) for resistance are shown for the top three drugs by LOR when data for more than three related drugs was available. Drug class AMG refers to aminoglycoside and individual drug abbreviations are available in Supplementary Data 7.

**Table 2 | 14 gene alleles predicted to be associated with resistance against specific drug classes for individual species**

| Species | Drug Class | Accession ID (Gene) | Predicted Gene Product | Muts.[a] | Resistant/Susceptible | LORs |
|---|---|---|---|---|---|---|
| Acinetobacter baumannii | quinolone | WP_000586912.1 (lptF) | Lipopolysaccharide export system permease protein | – | CIP = 795/166 LVX = 731/143 | CIP = 3.0 LVX = 2.5 |
| Acinetobacter baumannii | tetracycline | ADXO3353.1 (bccA) | Biotin carboxylase | – | TET = 107/0 MIN = 86/20 | TET = 7.4 MIN = 4.0 |
| Campylobacter coli | quinolone | AHK72934.1 (rny)[b] | Ribonuclease Y | d1-86, V87M | NAL = 41/16 CIP = 39/18 | NAL = 2.9 CIP = 2.6 |
| Escherichia coli | beta-lactam | WP_000811566.1 (yifN) | Putative uncharacterized protein, DUF147I | – | FOX = 46/0 CTT = 4/15 CXM = 246/136 | FOX = 9.4 CTT = 4.5 CXM = 3.2 |
| Escherichia coli | beta-lactam | WP_000118520.1 (sugE) | Small multidrug resistance (SMR) efflux transporter | T37A, M85A, A88L, A91G, L95A, +13 | FOX = 40/0 CAZ = 101/1 CRO = 85/0 | FOX = 9.0 CAZ = 7.7 CRO = 7.5 |
| Escherichia coli | beta-lactam | WP_001588947.1 (frdD)[c] | Fumarate reductase subunit D | V111D | AMC = 19/1 AMP = 14/0 CXM = 12/1 | AMC = 4.7 AMP = 4.5 CXM = 4.4 |
| Escherichia coli | quinolone | WP_000228346.1 (cycA)[c] | D-serine, D-alanine, glycine transporter | – | LVX = 251/35 CIP = 573/468 NAL = 41/17 | LVX = 4.0 CIP = 3.0 NAL = 0.7 |
| Klebsiella pneumoniae | beta-lactam | WP_004183775.1 | Cold shock protein of CSP family | V54A, H55L, A57T, Q62P, +2 | TIM = 16/2 AMP = 107/0 CEF = 13/0 | TIM = 7.0 AMP = 6.8 CEF = 4.1 |
| Salmonella enterica | beta-lactam | WP_001221666.1 (blc) | Lipocalin Blc | L49F, S98D, S175P, +10 | CRO = 302/0 AMC = 301/0 CTF = 297/3 | CRO = 13.6 AMC = 12.1 CTF = 10.9 |
| Salmonella enterica | beta-lactam | WP_000118520.1 (sugE) | Small multidrug resistance (SMR) efflux transporter | T37A, A104T, +9 | CRO = 304/0 AMC = 303/0 CTF = 299/3 | CRO = 13.7 AMC = 12.1 CTF = 11.0 |
| Staphylococcus aureus | beta-lactam | WP_000872606.1 (maoC) | MaoC domain protein | – | OXA = 29/3 MET = 201/4 FOX = 713/0 | OXA = 15.3 MET = 14.5 FOX = 13.1 |
| Staphylococcus aureus | beta-lactam | WP_000958858.1 | Glycerophos-phoryldiester phosphodiesterase | – | OXA = 29/3 FOX = 775/0 MET = 201/3 | OXA = 15.3 FOX = 15.2 MET = 14.8 |
| Streptococcus pneumoniae | beta-lactam | WP_000248982.1 | Cell wall surface anchor family protein | V66I, D111E, S160G, E172K | AMX = 13/2 MEM = 15/0 CXM = 15/0 | AMX = 13.1 MEM = 10.1 CXM = 9.7 |
| Streptococcus pneumoniae | beta-lactam | WP_000203066.1 (gpo) | Glutathione peroxidase | A80T, S132G | AMX = 13/3 MEM = 16/0 CXM = 16/0 | AMX = 12.7 MEM = 10.6 CXM = 10.3 |

Accession IDs are provided for the exact sequence of each allele (RefSeq when possible, GenBank otherwise), along with gene names when available, gene products, and mutations with respect to the most commonly observed allele. The number of resistant/susceptible genomes and log₂ odds ratios (LORs) for resistance are shown for the top three drugs by LOR when data was available. Abbreviations are available in Supplementary Data 7.

[a]When more than four mutations are detected, only mutations with BLOSUM62 score ≤0 are shown and the number of additional mutations is listed at the end. Full mutation sets are available in Supplementary Data 7. If no mutations are shown, the allele of interest is the most commonly observed allele.

[b]No exact variant was found on GenBank. AHK72934.1 refers to the most common allele of the gene, while the allele of interest contains the 86-amino acid N-terminal truncation d1-86.

[c]Selected for experimental validation.

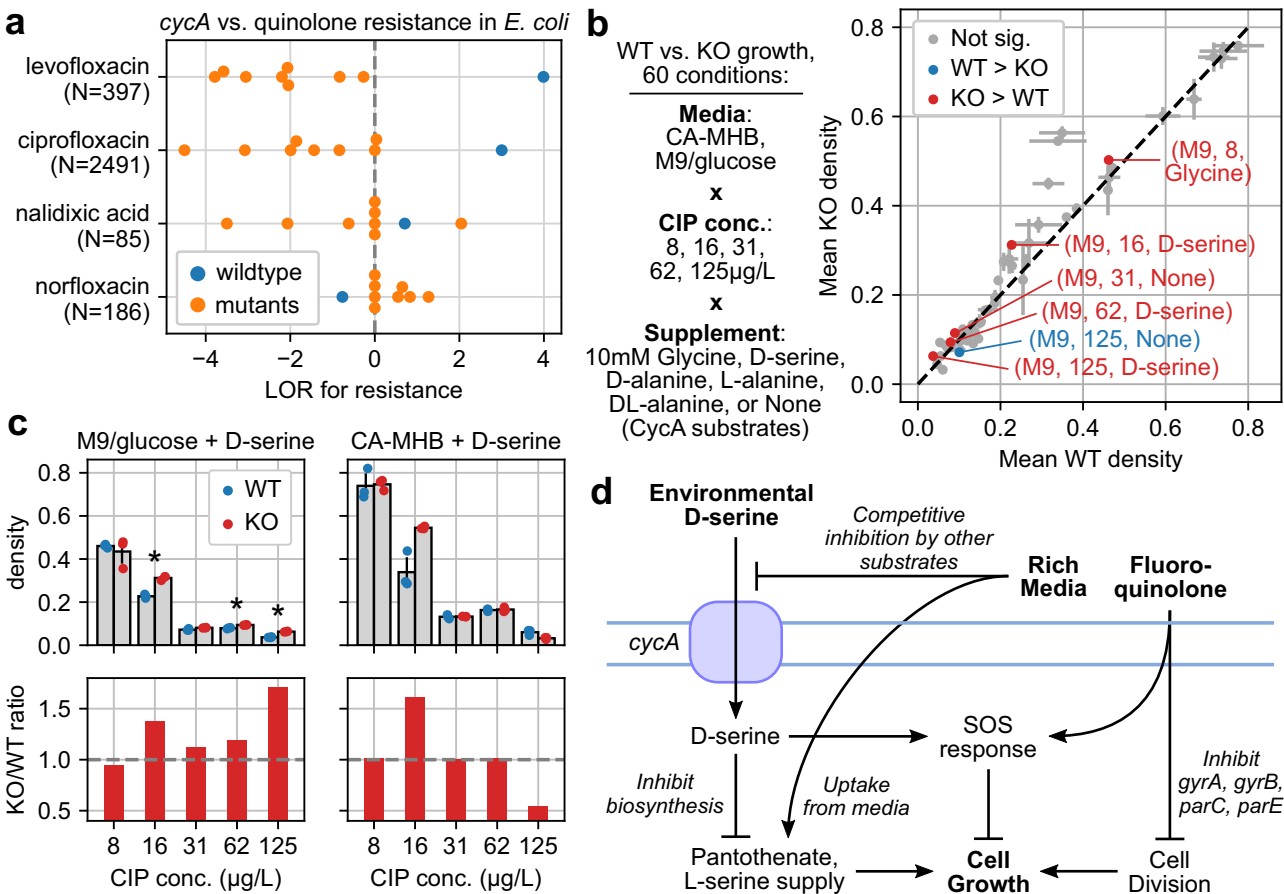

**Fig. 5 | D-serine-dependent impact of amino acid transporter CycA on quinolone resistance. a** Enrichment for resistant over susceptible genomes across observed *cycA* variants for four quinolone drugs, based on log₂ odds ratio (LOR). Number of genomes with AMR data is shown for each drug, and only variants observed in at least 100 genomes are shown. **b** Maximum cell density (OD600) achieved by *E. coli* BW25113 wildtype (WT) vs. *cycA* knockout (KO) across 60 conditions from combinations between media, ciprofloxacin (CIP) concentration, and CycA substrate supplementation. Error bars indicate standard deviations centered around means from biologically independent triplicates. Conditions where density differs significantly between WT and KO are labeled (n_WT=n_KO = 3, two-sided Welch

*t*-test, Benjamini–Hochberg correction, FDR < 0.05). **c** Absolute (above) and relative (below) maximum densities between WT and KO in conditions involving D-serine. Starred conditions correspond to the same significant differences from Fig. 5b. Error bars indicate standard deviations centered around means from biologically independent triplicates as in Fig. 5b for absolute densities. Relative densities show ratios between means from absolute densities. Significant differences between WT and KO are starred (*p* = 0.00040, 0.00344, and 0.00004 for CIP concentrations in M9 of 16, 62, and 125 μg/L, respectively). **d** Possible model for interactions between D-serine, *cycA*, media, and fluoroquinolones consistent with the observed conditional resistance conferred by *cycA* KO.

that *frdD* mutations yielding the V111D substitution requires *ampC* to confer beta-lactam resistance and is unlikely to contribute to resistance through any *ampC*-independent mechanism.

## Discussion

The exponential growth of publicly-available bacterial genome sequences and resistance metadata provides valuable opportunities for applying statistical methods to elucidate the genetics of AMR at a global scale. By applying workflows in pangenome construction, systematic AMR gene annotation, and machine learning to 27,155 genomes, 12 species, and 176,911 SIR phenotypes, we have characterized the current interspecies distribution of known AMR genes and demonstrated the broad capability of ML to carry out GWAS analyses of AMR, surpassing contemporary statistical methods at recovering known AMR genes. From the most accurate ML models, we have identified 142 AMR genetic feature candidates, two of which we have experimentally verified to impact resistance in *E. coli*. Many of these results depend on the reliable recovery of rare biological features and events and were enabled by the scale of this analysis, operating on the largest internally-consistent AMR phenotype dataset known to the authors at the time of publication.

First, an analysis of 6332 known AMR genes revealed 925 (14.6%) genes to be present in multiple species, which tended to be plasmid-encoded over chromosomally-encoded in a function-dependent manner. This result is consistent with previous studies finding plasmids frequently responsible for cross-species[46] and cross-genera[47] transfer of AMR genes in clinical environments. However, we also find that AMR gene transfer is much rarer between species differing at higher phylogenetic ranks, with just 8 AMR genes observed in at least 10 genomes outside their main phylogenetic class. It has been suggested that transfer of AMR genes between unrelated species such as between gram-positive (GP) and gram-negative (GN) species is rare but possible, having been inferred for tetracycline resistance proteins, *ermB*, *aph(3')*[48], and observed for some beta-lactamases[49].

A case study of *blaTEM* revealed one variant, TEM-116, to be present in the GP species *S. aureus* and found among *S. aureus* strains with the highest levels of resistance to beta-lactams. TEM-116 was observed on three plasmids of which plasmid NZ_AJ437107.1 was observed in both GP and GN strains, suggesting a potential route of transfer. While *blaTEM* was only recently reported in *S. aureus*[50], the *S. aureus* genomes we identified to harbor TEM-116 were isolated as early as 2009, suggesting that this transfer may be a much older phenomenon.

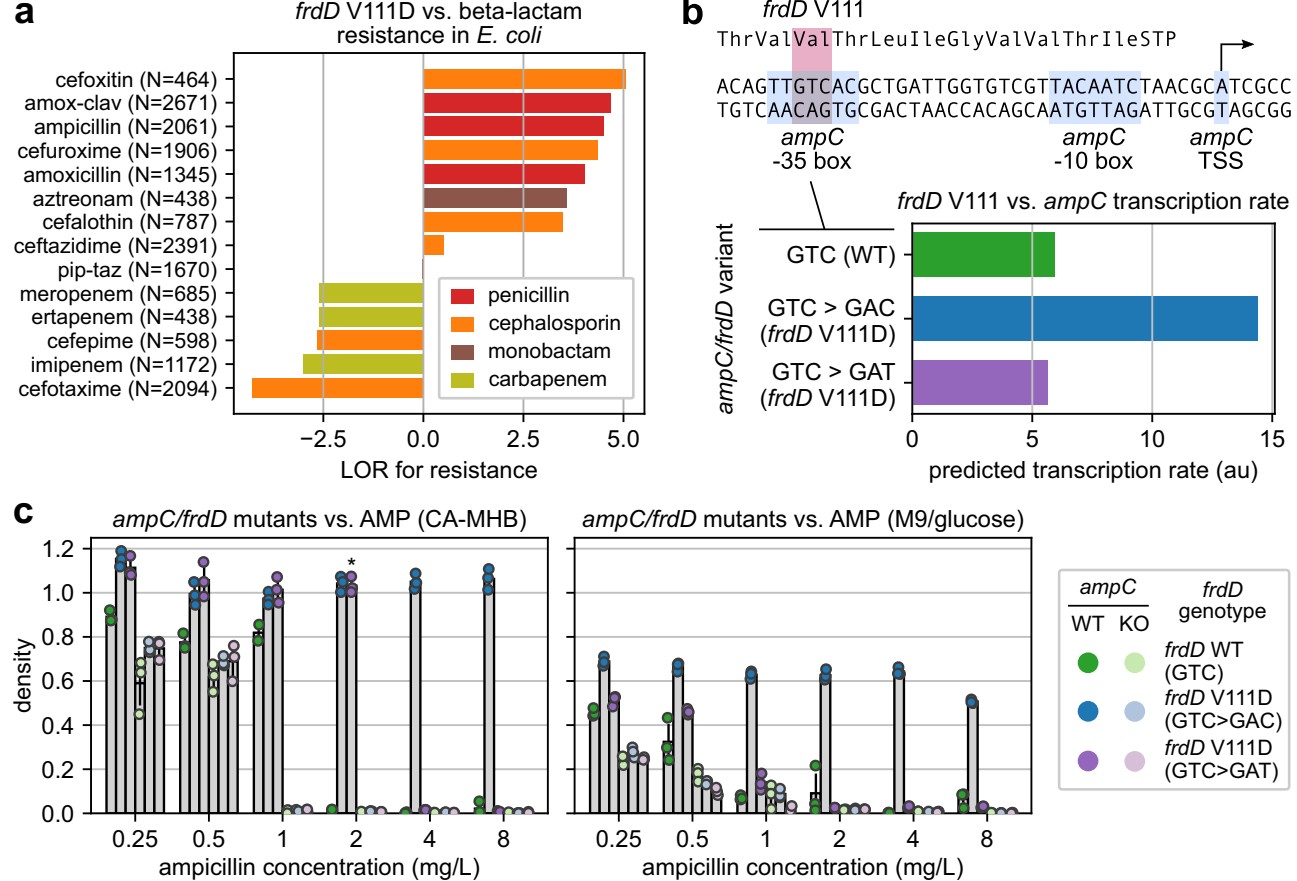

**Fig. 6 | *ampC*-dependent beta-lactam resistance conferred by the V111D substitution in fumarate reductase subunit FrdD. a** Enrichment for resistant over susceptible genomes from the presence of *frdD* mutations resulting in the V111D substitution across 14 beta-lactam drugs, based on log₂ odds ratio (LOR). Number of genomes with AMR data is shown for each drug. **b** Overlap between the *frdD* coding region and *ampC* promoter in *E. coli* BW25113 and predicted *ampC* transcription rates for various *frdD* mutations affecting V111. **c** Maximum cell density

(OD600) achieved by *ampC* and *frdD* mutants under increasing ampicillin (AMP) concentrations in rich (CA-MHB) or minimal (M9) media. Error bars indicate standard deviations centered around means from biologically independent triplicates. Six genotypes were tested, from combinations between three possible codons in *frdD* affecting V111 in either the BW25113 wildtype (WT) or corresponding *ampC* knockout (KO) strain. Starred case indicates growth was not observed until at least 8 h after inoculation for all replicates.

Further analysis of isolation dates may enable the reconstruction of timelines for the spread of multispecies AMR genes, and help identify whether specific species act as reservoirs for enabling interspecies AMR gene transfer. Given these observations, the transfer of AMR genes between GP and GN strains should be treated as a potential significant contributor to the spread of resistance, and large-scale analyses will likely be necessary to continue capturing these rare events of AMR gene transfer between unrelated strains.

We then developed a ML workflow for identifying AMR-associated genetic features by optimizing models for both phenotype prediction accuracy and biological relevance, with the latter quantified through a score based on the rankings of known AMR genes among a model's predictive features. Among cases with over 1000 SIRs and many known AMR genes, the GWAS score was found to generalize well to unseen known AMR genes, based on experiments randomly hiding known AMR genes from GWAS score calculation. Next, in a systematic analysis of four HPs, we found that the optimal ensemble size rarely exceeded 50 estimators, much lower than previous works and suggesting that smaller, more computationally efficient models are sufficient at this scale. Feature subsampling was also confirmed to improve the recovery of known AMR genes without compromising accuracy in a majority of cases. However, HP optimization offered only modest improvements over using fixed HPs, in contrast to a previous GWAS analysis with neural networks on datasets of comparable scale which found HP

optimization to significantly improve performance[51]. Additional analyses are needed to explore the practical limits of HP optimization for SVMs and AMR genetics, especially regarding how the competing metrics of accuracy and biological relevance may limit performance gains.

Applying this approach to 127 species-drug cases yielded models that were both accurate and reliably recovered known AMR genes. Compared with Pyseer and Fisher's exact test, SVM ensembles recovered nearly twice as many known AMR genes in total and near supersets of what could be recovered by the other methods. AMR genes recovered by multiple methods were concentrated among the top 3 features of the corresponding SVM ensemble, suggesting that only genes with the strongest statistical signals are reliably recovered by Pyseer or Fisher's exact test. Additionally, these results confirmed at a larger scale that accuracy is necessary but does not guarantee biological relevance[25]. This is likely due to strong population structure resulting from the clonal nature of bacteria, which can lead to significant correlations between causal and hitchhiker mutations that are difficult to distinguish using statistical approaches[52]. Though the SVM approach presented here does not directly address population structure, it was previously shown that attempting to do so through weighted sampling of strain subtypes did not improve AMR gene recovery over random sampling[25]. While other tools with different approaches for addressing population stratification may outperform

those tested here, ML approaches can supplement AMR gene recovery without requiring the often computationally expensive task of defining population structure in advance. These results suggest that even small SVM ensembles with limited HP tuning can reliably carry out GWAS analyses for AMR genetics.

Analysis of the most accurate models yielded 142 AMR gene candidates, two of which were selected for experimental validation: amino acid transporter CycA vs. quinolone resistance, and fumarate reductase subunit FrdD vs. beta-lactam resistance. Testing the effects of *cycA* KO vs. WT in *E. coli* in various environments, we found the *cycA* KO confers modest but significant, conditional resistance against CIP, specifically in minimal media supplemented with the CycA substrate D-serine. One possible explanation of this result involves the 1) toxicity of D-serine[38], which may be mitigated by reducing uptake through loss of CycA or competitive inhibition by other CycA substrates in rich media[37], and 2) the SOS response, which is triggered to differing extents by both D-serine[39] and fluoroquinolones[40] and may result in an SOS response induction no longer optimal to either stress and ultimately greater CIP susceptibility[41]. As neither CycA nor D-serine directly influence CIP's mechanism of action yet ultimately impact growth under CIP exposure, these results provide an example of separate environmental stresses and related genes measurably influencing AMR. Further investigation into D-serine may continue to shed light onto clinically relevant conditional resistance, as D-serine has also been shown to sensitize *S. aureus* to various beta-lactams[53] and concentrations of D-serine similar to those tested here may be encountered in various host environments[39,54]. CycA substrate glycine has also been shown to sensitize serum-resistant *E. coli* but not Δ*cycA* mutants[55], suggesting that CycA plays a role in multiple cases of environment-dependent resistance.

Similar experiments on the effect of the *frdD* V111D substitution on beta-lactam resistance revealed that resistance was conferred only in the presence of beta-lactamase encoding *ampC*, and comparing synonymous codons found this was likely due to overexpression of *ampC* as its promoter overlaps with the *frdD* open reading frame[42]. This substitution was previously associated with ampicillin resistance in *E. coli* induction experiments but not discussed in relation to *ampC*[56], while a similar synonymous mutation yielding V117V was previously shown to be enriched in amoxicillin resistant *E. coli* and attributed to *ampC* overexpression[57]. Given the prevalence of overlapping genes in bacterial genomes[58], future GWAS analyses that identify candidate genes associated with a phenotype will benefit from incorporating the genetic context of such genes into their analysis.

Overall, combining pangenomics, systematic gene annotation, and ML provides a workflow for efficiently uncovering patterns of known and candidate AMR genes at the scale of 10,000 s of genomes with potentially greater reliability than contemporary statistical methods such as Pyseer or Fisher's exact test. The flexibility of ML provides numerous opportunities to continue improving this workflow, such as the direct integration of the known AMR gene recovery into the loss function, use of different model architectures beyond SVMs, input of additional genetic feature types such as SNPs, or benchmarking against other phenotypes beyond AMR. As the number of genome sequences continues to grow, periodic updates to this analysis with improved techniques will likely be able to capitalize on the benefits of scale and steadily deepen our understanding of AMR across the phylogenetic tree. Continued development is necessary to bring ML into the current GWAS toolbox when mining sequencing data for previously unknown genetic determinants underlying complex phenotypes.

## Methods

### Genome selection

An initial set of genomes was taken from the PATRIC database RELEASE_NOTES (ftp.patricbrc.org/RELEASE_NOTES/, 2021-07-21) and filtered by taxon ID to 12 species among the ESKAPEE pathogens or WHO global priority pathogens released in 2017. For each species, genomes were filtered to those meeting four criteria[26]: 1) genome status is "WGS" or "Complete", 2) number of contigs is within 2.5 times the median number of contigs across all assemblies for that species, 3) number of annotated CDSs is within 3 standard deviations of the mean, and 4) total genome length is within 3 standard deviations of the mean. Genomes were further filtered for those having EvalCon fine consistency[59] of at least 87%[60]. Drugs with at least 100 experimental antimicrobial (AMR) measurements were identified per species, and genomes were filtered to those with data for at least one of those drugs. Selected PATRIC Genome IDs are available in Supplementary Data 1. Genome MLST subtypes were annotated using mlst v2.18.0 (https://github.com/tseemann/mlst), and BioProject IDs were taken directly from metadata available on PATRIC, with distributions available in Supplementary Data 2.

### Pangenome construction and genetic feature identification

For each species, genes (protein sequence clusters), alleles (protein sequence variants), and ORF-flanking sequence variants were identified using a sequencing clustering approach[26]. All unique protein sequences across all genome assemblies (as annotated by PATRIC) were clustered using CD-HIT v4.6 with minimum identity 80% and minimum alignment length 80%[29]. Each cluster was treated as a gene, cluster members as alleles, and the 300 bp upstream of the start codon of each occurrence of the gene as 5′ variants (and analogously the 300 bp downstream of stop codons as 3′ variants). 5′/3′ variants of a gene were identified by locating all occurrences of all alleles across all genome assemblies and extracting the 300 bp directly flanking those occurrences. Cases in which a 300 bp flanking region is interrupted by a contig break were ignored (for ML purposes, such genomes were treated as not having any specific 5′/3′ variant). A similar clustering analysis was conducted for non-coding features (annotated by PATRIC as "transcript", "tRNA", "rRNA", or "misc_binding") to identify noncoding feature clusters and nucleotide sequence variants, using CD-HIT-EST v4.6 with the same parameters[29]. The species-wide genetic variation covered by these six feature types was represented as a binary matrix based on presence/absence calls of each feature for each genome.

### Processing antimicrobial resistance phenotypes and SIR phenotype inference from MICs

Species-drug cases with at least 100 experimental susceptible-intermediate-resistant (SIR) phenotypes or minimum inhibitory concentration (MIC) measurements among selected genomes were identified. For each pair, the most common SIR testing standard (i.e. CLSI, EUCAST) was identified from either the metadata or manual curation of BioProject accession IDs (Supplementary Data 2), referred to as the species-drug case's primary standard.

MIC values were mapped to SIRs by first filtering MICs for exact values mg/L values (opposed to bounded MICs) derived from one of the following laboratory typing methods: agar_dilution, agar_dilution_or_etest, bd_phoenix, bd_phoenix_and_etest, broth_microdilution, etest, mic, mic broth microdilution, liofilchem, sensititre, vitek_2. For each species-drug case, MIC-SIR mappings were generated for all MIC-SIR value pairs reported in at least three genomes under the primary standard. Ambiguous mappings (MIC value mapped to multiple SIRs) and inconsistent sets of mappings (instances where a susceptible MIC is greater than an intermediate or resistant MIC, or where a resistant MIC is less than an intermediate or susceptible MIC) were removed. A MIC-to-SIR inference scheme was developed as follows:

- Exact MICs: Mapped directly to the corresponding SIR if possible.
- Upper bounded and unmapped exact MICs: If MIC ≤ largest MIC mapped to susceptible, it is mapped to susceptible.

- Lower bounded and unmapped exact MICs: If MIC ≥ smallest MIC mapped to resistant, it is mapped to resistant.
- Other unmapped exact MICs: If the MIC value is within the range of MICs mapped to intermediate, it is mapped to intermediate.

MICs used and MIC-SIR mappings are available in Supplementary Data 2. Reported and inferred SIRs were combined for subsequent analyses. For genomes with multiple conflicting SIRs for a single drug (i.e. measured by different methods), the most common SIR across all methods and inferences was selected, with directly reported SIRs breaking ties and perfect ties ignored. The final set of SIRs are available in Supplementary Data 2. SIRs were binarized into susceptible and non-susceptible by converting "susceptible", "susceptible-dose dependent", and "non-resistant" to 0 s, and "resistant", "intermediate", "non-susceptible" and "IS" to 1 s for subsequent analyses.

### Identification and classification of known AMR genes

All protein sequences for each species were annotated using RGI v5.2.0 with CARD ontology v3.1.3[27]. To link AMR genes identified by RGI to specific drugs, a directed graph was constructed from the CARD ontology using ARO accession IDs as nodes and adding directed edge "U ->V" whenever:

- U corresponds to a gene and U has the relationship "is_a" to V.
- U corresponds to a drug and V has the relationship "is_a" to U.
- U has the relationship "part_of", "regulates", "confers_resistance_to_antibiotic", or "confers_resistance_to_drug_class" to V.
- V has the relationship "has_part" to U.

A gene was labeled as conferring resistance to a drug if there exists a path from the gene's node to the drug's node in this graph. 23S rRNAs, 16S rRNAs, and 50S rRNAs were manually identified from PATRIC text annotations of noncoding features and were similarly linked to specific drugs using the graph, starting from nodes ARO:3000336, ARO:3003211, and ARO:3005003, respectively.

Additional drug-specific AMR genes were identified from PATRIC text annotations. Sequences with annotations containing a drug name or identical to the annotation of an RGI-identified AMR gene were identified and manually curated for probable known AMR genes (curated annotations are available in Supplementary Data 3). For machine learning purposes, all features associated with a gene cluster containing a sequence linked to resistance for a drug by RGI or PATRIC text annotation were treated as known AMR features. The distribution of AMR phenotypes for genomes carrying features annotated as known AMR is available in the Supplementary Analysis (Supplementary Fig. 14).

### AMR gene cross-species comparison, location prediction, and TEM beta-lactamase analysis

All alleles of identified AMR genes across all species were combined, de-duplicated, and re-clustered with the same CD-HIT parameters for cross-species analysis. Gene-level functional annotations for re-clustered AMR genes were inherited from corresponding allele-level annotations from RGI and PATRIC. AMR gene functional categories were assigned based on RGI annotations when available and PATRIC annotations otherwise, with categories occurring less than 50 times grouped as "other" (Supplementary Data 3).

All contigs from all assemblies were labeled as plasmid or chromosomal based on de novo predictions from PlasFlow v1.1 on default settings[30]. Contigs were also mapped to known plasmids in PLSDB version 2021_06_23[61] using MASH v2.3 with minimum shared kmers 500/1000[62]. For a given AMR gene cluster, each instance of each allele was assigned a location based on the PlasFlow prediction for the contig containing that instance (chromosome, plasmid, unassigned). The overall location of a gene or allele was assigned as 1) "chromosome" if >90% of instances were chromosomal, 2) "plasmid" if >90% of

instances were plasmid, 3) "chromosome-leaning" if >50% of instances were chromosomal, 4) "plasmid-leaning" if >50% of instances were plasmid, and 5) "ambiguous" otherwise (Supplementary Data 3).

Complete TEM-family beta-lactamases (*blaTEMs*) were identified by filtering all AMR alleles for mention of "TEM" in the RGI or PATRIC annotation, and for length at least 272aa (95% length of TEM-1). Mutations were called from pairwise global alignment of each allele to TEM-1 using the Biopython Align module[63] with scores match = 1, mismatch = −3, open_gap_score = −5, and extend_gap_score = −2. N/C-terminal deletions were ignored. Known TEM variants were identified based on exact matches in the CARD database. Presence of specific *blaTEM* plasmids in individual genomes was determined by filtering the previous MASH results against PLSDB for mappings with MASH distance <0.025 (Supplementary Data 4). For the *S. aureus* analysis, cefoxitin was identified as the only drug for which MIC data was available among *blaTEM*-carrying *S. aureus* genomes. Known AMR genes among *S. aureus* genomes with cefoxitin MIC data were identified from exact matches to entries related to beta-lactams in the CARD database.

### Implementation, evaluation, and hyperparameter optimization of SVM ensembles

For each species-drug case, SVM ensembles were trained to classify genomes as susceptible or non-susceptible (intermediate or resistant, referred to as "resistant") based on the species' genetic feature presence/absence matrix. To accelerate training, feature count was reduced in three stages: 1) features present or missing in less than 3 genomes were removed, 2) perfectly correlated features were merged, and 3) remaining features were sorted by log odds ratio (LOR) for resistant genomes, and features with the 25,000 highest and 25,000 lowest LORs were retained (variations to this input filter were tested in the Supplementary Analysis, Supplementary Fig. 15, Supplementary Data 5). SVM ensembles were implemented using scikit-learn v1.0.1 classes LinearSVC and BaggingClassifier[64], with square hinge loss (loss = 'squared_hinge') weighted by class frequency (class_weight = 'balanced') to address class imbalance issues and L1 regularization (penalty = 'l1') to enforce sparsity in feature selection.

Model performance was evaluated in 5-fold cross validation experiments. Phenotype prediction accuracy was scored as the mean Matthews correlation coefficient (MCC) on the test set. Biological relevance was scored using the following equation:

$$GWASScore = \sum_{r \in known} 0.5^{(r-1)/10} \qquad (1)$$

where r corresponds to the ranks of known AMR features associated with the specific drug when sorted by feature importance, with r = 1 corresponding to the highest feature importance. Feature importance was computed as the absolute value of the mean of the feature's coefficients across all SVMs in the ensemble with access to the feature (i.e. selected during feature subsampling). For ties, the average rank was assigned to all tied features.

Hyperparameter (HP) ranges for SVM ensembles were first evaluated on 10 test species-drug cases (Supplementary Table 4). These were selected by first filtering for cases with substantial available data (at least 1000 SIRs and 50 known AMR genes), then sampling for even representation of drug classes and species phylogenetic classes. 256 HP combinations from four HPs were tested for each test case: number of estimators per ensemble, fraction of samples per estimator (with replacement), fraction of features per estimator (without replacement), and the SVM regularization term C, corresponding to parameters 'n_estimators', 'max_samples', and 'max_features' in BaggingClassifier and 'C' in LinearSVC, respectively. For each test case, the highest MCC (mean test set MCC from 5-fold CV) and GWAS scores

were computed across models for all HP combinations. The smallest subset of HP combinations was identified such that the best scores in the subset were within 90% of the best scores across all combinations, across all test cases. The initial and reduced HP ranges are available in Supplementary Table 6.

SVM ensembles were trained for each HP combination in the reduced HP set across 127 species-drug cases with at least 100 SIRs, 10 known AMR genes, and minority phenotype frequency >5%. For each case, the optimal HP set was selected by ranking all HP combinations by either MCC or GWAS score, taking the average of the two ranks as the HP set's overall rank, and selecting the HP set with the highest overall rank. Final model performance and selected HPs are available in Supplementary Data 5.

### Comparison between SVM ensembles, Pyseer, and Fisher's exact test at recovering known AMR genes

For each of the 127 species-drug cases tested, the top 10, 20, and 50 genetic features associated with the AMR phenotype were identified for each GWAS approach. For SVM ensembles, features were sorted by feature importance as previously defined. For Fisher's exact test, the test was applied between each genetic feature and the binary AMR phenotype (susceptible/non-susceptible), and features were sorted by $p$-value. For Pyseer[21], distances were first computed between each pair of genomes using MASH v2.3 with default parameters[62]. Pyseer v1.3.10 was then run using binary AMR phenotypes for the --phenotype parameter, genetic feature presence/absence calls for the --pres parameter, MASH distances for the --distances parameter, and all other parameters default, and features were sorted by population structure-adjusted $p$-value "lrt-pvalue".

### Identification of candidate antimicrobial resistance determinants

The models for each of the 127 species-drug cases were filtered down to those achieving mean test MCC > 0.8 during 5-fold cross validation. From each remaining model, the top 10 features by feature weight absolute value were identified, and filtered for those that 1) were not already known AMR genes, 2) occurred in at least 10 genomes with SIR data for the corresponding drug, and 3) had positive feature weight and LOR for resistance. Remaining feature-drug pairs were tested for whether the feature was significantly associated with resistance, applying Fisher's exact test for SIRs and Brunner-Munzel tests for MICs. Tests were applied for the specific drug and drugs of the same class for which at least 5 SIRs (for Fisher's exact) or 5 MICs (for Brunner-Munzel) were available, and significance was determined at FWER < 0.05 with Bonferroni correction (2008 Fisher's exact tests and 1393 Brunner-Munzel tests were conducted, with significance thresholds of $p < 2.5*10^{-5}$ and $p < 3.6*10^{-5}$, respectively). To evaluate co-occurrence with known AMR genes, AMR features found in predominantly in resistant strains were identified for each drug, defined as those occurring in at least 5 genomes of which at least 90% are resistant. Each candidate feature-drug pair was assigned a score based on the sum of 1) number of drugs with significant association based on SIR data, 2) number of drugs with significant association based on MIC, and 3) number of drugs for which at least one resistant genome with the feature does not also have any AMR features found predominantly in resistant strains. Candidate scores are available in Supplementary Data 7.

The top 10 features by score for each species-drug class pair were identified, yielding 142 candidates which were categorized by function as annotated by PATRIC and additionally by eggNOG-emapper v2.1.6-43[65]. Genes that were poorly annotated, related to mobile genetic elements (transposases, insertion elements, phage elements, integrases, plasmid maintenance), or known to be associated with a specific AMR mechanism for an unrelated drug class were their own categories, and the remaining genes were categorized as well-

characterized candidates. For perfectly correlated features, coding features were selected over noncoding features. For variant-level features, mutations were determined against the most common variant of the parent gene cluster using the Biopython Align module[63]. Interpretation of *E. coli* candidates was conducted using reference genomes U00096.3 [https://www.ncbi.nlm.nih.gov/nuccore/U00096.3] for K-12 MG1655 and NZ_CP009273.1 [https://www.ncbi.nlm.nih.gov/nuccore/NZ_CP009273.1] for BW25113.

### Generation of *frdD* and *cycA* E. coli mutants

*E. coli* BW25113 knockout mutants *ΔcycA* and *ΔampC* were taken from the Keio collection[36]. The *frdD* mutations referred to in this study were introduced into both BW25113 and *ΔampC* using a Cas9-assisted Lambda Red homologous recombination method. Golden gate assembly was first used to construct a plasmid vector harboring both Cas9 and lambda red recombinase genes under the control of an L-arabinose inducible promoter, a single guide RNA sequence, and a donor fragment generated by PCR which contained the desired mutation and around 200 bp flanking both sides of the Cas9 target cut site as directed by the guide RNA. After allowing the transformed cells to recover for 2 h at 30 °C, L-arabinose was added to the media and the cells were allowed to grow for 3–5 h at which time a portion of the culture was plated. Single colonies were screened using ARMS PCR and amplicons spanning the mutation site, generated with primers annealing to the genome upstream and downstream of the sequence of the donor fragment contained in the plasmid, were confirmed with Sanger sequencing. Confirmed isolates were cured of the plasmid by growth at 37 °C. Both of the *frdD* mutations that were introduced in this study fell within a guide RNA target sequence. Because Cas9 has a tolerance for some single base mismatches in the guide RNA, a second mismatch was engineered into the guide RNA so that the guide RNA had two mismatches with respect to the successfully mutated target sequence and only one tolerated mismatch with respect to the starting strain. In one case, an intermediary strain was first constructed in which all of the codons falling within the guide RNA target sequence were switched to synonymous ones maximizing the base changes. A second round of Cas9-assisted Lambda Red homologous recombination was then used to restore those codons to their original sequences and introduce the desired mutation at the same time.

### Cell growth conditions and measurements

Two media were used for both *cycA* and *frdD* experiments: 1) Mueller–Hinton Broth (Sigma-Aldrich, SKU: 70192-500 G) supplemented with 49 mM $MgCl_2$ and 69 mM $CaCl_2$, and 2) M9 minimal medium (47.8 mM $Na_2HPO_4$, 22 mM $KH_2PO_4$, 8.6 mM NaCl, 18.7 mM $NH_4Cl$, 2 mM $MgSO_4$, 0.1 mM $CaCl_2$) supplemented with 2 g/L glucose. For *cycA* experiments, media were also supplemented with either 10 mM glycine, D-serine, D-alanine, L-alanine, DL-alanine (50:50 mixture of D- and L-alanine) or nothing, for a total of 12 possible supplemented media.

Cell densities for strain-media-antibiotic combinations were measured in biological triplicates as follows: Fresh culture samples were prepared (OD600 = -0.05) in each relevant media. Sample solutions were loaded to Costar flat-bottom 96-well plates (Corning, catalog no. 3370), with antibiotics added to varying concentrations (8, 16, 31, 62, or 125 μg/L ciprofloxacin for *cycA* experiments, and 0.25, 0.5, 1, 2, 4, or 8 mg/L ampicillin for *frdD* experiments). Plates were incubated in a microplate reader (Tecan Infinite200 PRO) with shaking at 37 °C, and ODs were read every 15 min. Maximum cell density for each condition and replicate was calculated as the maximum OD600 over 12 h after inoculation minus the minimum OD600 observed for the corresponding media without inoculum. Significant differences in cell density between pairs of strains or conditions was determined with Welch $t$-tests (FDR < 0.05, Benjamini–Hochberg correction).

## Statistics & reproducibility

No statistical method was used to predetermine sample size. Sample size was limited to the set of all publicly available genomes with AMR metadata passing quality thresholds (see Methods). Genomes failing any of five assembly quality thresholds were excluded from the analysis (see Methods). No exclusions were made in validation experiments. AMR validation experiments were conducted with biological triplicates and combinatorial condition design, and all attempts at replication were successful. The experiments were not randomized. Experiments were designed to test hypotheses generated by statistical analysis regarding the effect of specific genetic alterations and environments on AMR, and all possible combinations of conditions were tested. This is in contrast to a broader screening experiment for which comprehensive testing would be impossible and randomization would be required. The Investigators were not blinded to allocation during experiments and outcome assessment. Allocating all combinations of *E. coli* mutants/conditions to different investigators was infeasible, and not blinding is unlikely to interfere with the analysis or alter the results.

## Reporting summary

Further information on research design is available in the Nature Portfolio Reporting Summary linked to this article.

## Data availability

All genomes used in this study are publicly available on the PATRIC (now BV-BRC, https://www.bv-brc.org/) database. Accession IDs are available in Supplementary Data 1. Culture density data generated from validation experiments are available in Supplementary Data 8. Additional databases used during analysis are CARD database ontology v3.1.3 (https://card.mcmaster.ca/) and PLSDB v2021_06_23 (https://ccb-microbe.cs.uni-saarland.de/plsdb/). Processed data types used to generate figures and tables are available within the Supplementary Information and Source Data. Source data are provided as a Source Data file. Source data are provided with this paper.

## Code availability

Representative code for analyses in this study are available at https://github.com/jhyun95/pangenomix/releases/tag/1.0.0.

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

## Acknowledgements

This research was supported by a grant from the National Institute of Allergy and Infectious Diseases (AI124316, awarded to J.M.M. and B.O.P., https://www.niaid.nih.gov/). This research was also supported by a grant from the National Institutes of Health (T32GM8806, awarded to J.C.H., https://www.nih.gov/). The funders had no role in study design, data collection and analysis, decision to publish, or preparation of the manuscript.

## Author contributions

J.C.H., J.M.M., and B.O.P. contributed project Conceptualization. J.C.H. developed the computational Methodology and Software, and conducted the Data Curation, Investigation, Formal Analysis, Validation, and Visualization of results. R.S. developed the Methodology and conducted the Investigation for generating *E. coli* mutants. Y.H. developed the Methodology and conducted the Investigation for measuring growth of *E. coli* mutants in various conditions. J.M.M. and B.O.P. contributed to Funding Acquisition and Project Administration, and provided Supervision and Resources. J.C.H. prepared the Original Draft and all authors were involved in Review & Editing.

 

## Competing interests

The authors declare no competing interests.
