## [Peer Review File · Nature Communications]

REVIEWER COMMENTS

Reviewer #1 (Remarks to the Author):

Global pathogenomic analysis identifies known and novel genetic antimicrobial resistance determinants in twelve species

Summary:

Hyun et al present a novel machine learning approach to identify resistance determinant by looking at a large number of public genomes with matched AMR phenotypes. They also describe the diversity and breadth of AMR genes within the dataset and across species/genera/class.

This manuscript was well-written with excellent figures.

Overall I was disappointed that there wasn't more effort to compare with existing bacterial GWAS studies and tools, and it was unclear if common problems in bacterial GWAS (like population structure) were addressed.

General comments:

The authors mention AMR prediction but not existing GWAS approaches/studies in bacterial genomics in the introduction. This should be addressed.

How much variation is there in the genomes the authors acquired from PATRIC? Are they both genetically diverse and well sampled (without specific bias from a single study, and with sufficient sensitive/resistant strains, for example?). This has implications on the conclusions for AMR sharing between species and predictive power for antimicrobial resistance.

Generally the section on blaTEM in the results was a bit abrupt and did not seem to add to the overall aim of the manuscript, and as such could be relegated to supplementary.

In the section starting on line 149, where the authors "...train models for both accuracy at predicting AMR phenotypes and biological relevance..." could the authors pick a few relevant examples where the model did well at predicting phenotype/known AMR genes? E.g. meropenem resistance based on blaKPC. At the moment I find it difficult to see solid examples.

I generally found it a bit odd to compare the model to Fisher's exact test. Can the authors explain why they chose this? I wouldn't say it's commonly used, and there are many existing microbial GWAS tools available that could have been benchmarked against (Pyseer, bugwas, dbgwas).

Specific comments:

Line 74-75: Do the authors end up double-dipping certain variants if genes overlap and the authors are also taking 300 bp upstream/downstream? How does this affect the analysis?

Line 138: Could the authors use blast (or something like BRIG) to determine from the 28 whole genomes which of these plasmids are more likely, rather than focusing only on the TEM contig? Are the genomes clonally related or is this more likely HGT?

Line 146: I think it's worth mentioning that the authors have only n=4 samples here with all three genes. Are these independent genomes or is there an unobserved lineage effect?

Figure 2d: why are there three columns that all say blaTEM cluster? different TEM alleles?

line 163: "A feature was labeled as a known AMR genetic feature if its corresponding gene cluster was a known AMR gene for the drug of interest as annotated by RGI and PATRIC" - what about SNPs conferring resistance? e.g. rpoB/gyrA are not resistance genes, but SNPs in them can confer resistance.

Line 166: I am unclear on whether the authors trained and evaluated the model on the same data, or on different subsets. Could the authors explain this more, and their rationale?

Line 182: "accuracy did not guarantee biological relevance" Is this because of strong population structure? Does the model account for this at all? This is a well known problem with bacterial GWAS

Line 183: "Certain dataset parameters were weakly but significantly associated..." I don't see this in the S6c plot, what is significant here?

Line 190: "The top 50 features for each model are available in Dataset S5, and their sequences in Dataset S6." I may have misunderstood, but the authors have >100 models, so should the lists in these datasets be much longer?

Line 256: Were there any baseline growth differences between WT and delta-cycA? What do the results look like with only CIP in rich/minimal media, without supplementing substrates?

Line 261: it appears the cycA knockout is doing comparably/better to the WT - the authors haven't explicitly said this in the results here. What does this mean for the authors ML model, since it predicted the WT associated with quinolone resistance?

Line 297: did the authors sequence the strain after the experiment? Is it possible it acquired the mutation?

Line 426: "For each species, genomes were filtered to those meeting the four criteria previously described" please add the specifics for reproducibility

line 427: What is "fine consistency"? I haven't heard this term before

Line 437: How did the authors obtain 300bp upstream/downstream, and what happened if there was a contig break before this?

minor:

line 17: specify "multiple phylogenetic classes"

line 125: remove "which"

Line 162: remove "a"

Reviewer #2 (Remarks to the Author):

Hyun and colleagues present a study on AMR. In the paper, they track AMR genes across diverse genomes, build SVM models to identify features that are associated with AMR in these genomes, and experimentally verify that two mutations – a deletion in *cycA* and a V111D variant in *frdD* --have a role in resistance. Overall, I think the paper has novelty and is a nice contribution to the field. The science appears to be sound and most of my comments relate to clarifying the text.

Line 18 what do the authors mean by the traditional methods?

Line 21 The reference genome for the variant V111D needs to be declared here and elsewhere in the paper. I was originally thinking this was in MG1655, but it turned out not to be. Also, it is disorienting because the previous sentences are talking about resistance across the tree of life, and then this zooms in on an *E. coli* mutant, which isn't bad, it just needs clarification. Is BW25113 always your reference strain?

Based on Figure 1B, it looks like *M. tuberculosis* was excluded from the analysis even though it might have the most AST data. Was there a biological or strategic reason for this? It has a lot of SNP-based AMR, but also perhaps a lot to learn about.

Figure 1D MLSB should be defined in the text or figure legend

I think the authors need to be careful to fully qualify any statements comparing their SVM approach to GWAS. On line 362 in the discussion, they describe some of the more sophisticated tests for population stratification etc. While I do agree that doing this type of analysis would be inconvenient, the authors do not compare their approach with these more rigorous methods. That makes it difficult to say that it is better or outperforms GWAS.

It looks like some of the SVMs are based on the presence/absence of genes, is that also true for the GWAS comparison?

The authors ultimately ended up talking about amino acid level changes (particularly in the V111D mutant). But it was not clear to me how the upstream sequences were turned into features that allowed this to be detected by the SVM model. Were these detected after the fact?

Is frdD plasmid-borne? Do frdD and ampC always travel as a pair? If not, are there equivalent promoter mutations in upstream genes where ampC is located elsewhere that were found in the analysis?

I felt that nearly all of the description of the hyperparameter selection could be moved to the methods. It really slowed down the results section.

Line 91: What do they mean by best annotation?

Line 243: I got hung up on the term “coding variant” and how that relates to the deletion mutant that is ultimately reported.

Line 244 what species does the cycA refer to, and what is wild type

Why does figure 5C go: 8, 16, 31, 62, 125 instead of 8, 16, 32, 64, 128? Also on 5C, why isn't 31 significant? It seems odd that 16 ug/L has the greatest difference and then the trend almost disappears

Line 32 Reference should be updated to <https://pubmed.ncbi.nlm.nih.gov/34379107/>

Line 427 In the BV-BRC resource, fine consistency in genome quality comes from this paper, not checkM <https://pubmed.ncbi.nlm.nih.gov/31581946/>, although CheckM is often computed as well.

I was hoping for some discussion of how the authors ultimately decided to validate the cycA and frdD mutants, when there were hundreds to look at. Were there some that you couldn't work out? Based on my reading of this there were other features that ranked higher.

Line 262: I found this sentence to be awkward. “Across all conditions involving D-serine, while increasing CIP concentration reduced final density for both strains, the KO strain achieved higher densities for 16-125µg/L CIP, but only in M9 media and not in CA-MHB (Figure 5c)”.

272: “response that is adapted to neither stress” I had difficulty understanding this sentence.

I found the CycA section in the results to be generally difficult to read, but I think that the authors are suggesting a synergistic killing effect between cipro and D-Serine in M9 media (when L-serine can't compete with D-serine), which is fine, but what conditions in nature would ever cause this? Why would this ever evolve? Is there some other real-world compound that is transported by CycA that is the actual physiological culprit for the evolution of this conditional resistance mutant? Is there a scenario where D-serine is made, like in a macrophage?

Reviewer #3 (Remarks to the Author):

The work presented by Hyun et al. describes a workflow that combines pangenomics and machine learning (ML) to identify known and novel AMR genes. Using data retrieved from public repositories and encompassing 12 species spanning 27,155 genomes and 69 drugs. By applying this pipeline, they found that AMR gene transfer is mostly confined to related species. In this study, they identified 142 novel AMR gene candidates, two of which were studied experimentally and showed conditional resistance.

A few considerations and concerns are reported below:

Can the authors discuss what are the improvements (e.g. performance, genes found) obtained by the optimization of the hyperparameters, as done in this work, in comparison to the results found in Hyun et al. PLoS Comput. Biol. 16, e1007608 (2020)?

Line 15-17 abstract: "found AMR gene transfer mostly confined within related species" see also Line 95 Results and Line 330 onwards and discussion. The authors claim that they found AMR gene transfer, mentioning that their results were consistent with previous studies [REF 43 and 44]. I respectfully disagree with this assertion and with the claim that the authors can provide evidence for gene transfer in their work. In Ref 44 a collection of 2173 bacterial isolates from healthcare-associated infections from a single hospital over 18 months was performed. In that work, the authors performed a collection campaign where samples were collected in a specific environment and time following an appropriate protocol to track the appearance of plasmid transmission and to provide evidence of plasmid transfer. The work presented by Hyun et al. would require additional factors to translate their findings into transfer and to make inferences on gene transfer. Co-presence of AMR features does not imply transfer.

Line 155-157 Results and 524-526 Methods. The ML framework proposed by the authors classifies genomes into susceptible vs non-susceptible. To do this the authors group the isolates with intermediate phenotypes with those with resistant phenotypes. However, considering that intermediate values can range from a minimum value that is close to the max. value of susceptibility to an upper value

that is close to the minimum value of resistance, including the intermediate cases in the resistant group can create a bias (which could be a concern because the intermediates could be more closely related to susceptible samples). For example, Kim et al [REF 13] and others, when using machine learning and CLSI breakpoints to determine the antibiotic phenotype (sensitive versus resistant) from sequencing data, did not include the isolates with intermediate susceptibility in the analysis. Mislabeling observations in classification tasks can strongly affect the true outcome obtained by the learners.

Line 159-163 Results and Line 533-538 Methods. The authors propose a “GWAS score” as a metric to define the biological relevance of the AMR genes. Is this score normally used in GWAS studies? If yes, can the authors provide references; if not how the authors defined this score? Was it defined empirically? What is the reasoning behind it and the robustness to support its use? I could not find references in support of the robustness of this score and more importantly no benchmarking of the method.

The authors mention that with their method they can identify novel genes but if the GWAS score is defined as: “..a GWAS score based on the rankings of known AMR genes among a model’s predictive features” how can such a metric, which seems to strongly rely on what is known, correctly identify novel AMR genes (which can differ from the known ones)? The method seems biased towards the known AMR genes and might not fully address the novel genes.

Lines 166 Results and lines 542-543 Methods. How were the 10 “representative” species-drug cases chosen (lines 166 and 542-543)? What does “representative” means? Regarding the 127 species-drug cases, how are these 10 cases connected to them (how well do they represent the 127 cases?)?

In Figure S6B and on lines (170-172) the authors claim that a more aggressive feature subsampling increases the number of known AMR genes (GWAS score). However, the authors seem to not provide a statistical test in support of this claim.

When reducing the number of samples for the ensemble, does each base classifier have the same distribution of classes (i.e. resistant and susceptible) with respect to the full set of samples? Or the selection of samples is done randomly, hence no attention is paid to the class distribution? In this case, how does this affect the performance of the ensemble as a whole?

The authors are emphasizing the use of an ensemble approach; however, they do not demonstrate what the performance of a single SVM compared to the ensemble is like.

Lines 170-172 Results and lines 542-545 Methods. It is unclear how many known AMR features were present in each one of the base classifiers within the ensemble when the number of features in the

ensemble was lower than 100% (range 25%, 50%, 75% and 100%). Did the number of known AMR features affect the performance of the base classifiers? For example, in Figure S5, the models for *S. aureus* ceftazidime achieved a nearly perfect MCC, with almost no AMR features (GWAS score close to 0). Whereas, the models for *S. aureus* ciprofloxacin and erythromycin achieved a lower MCC with much more known AMR features present (GWAS score >4). The same thing happens with *S. enterica*. Shouldn't we expect the opposite (a higher GWAS score leading to higher performance)?

Lines 172-174 Results. It is unclear how the subset of HP combinations was chosen. The authors seem to select them by using thresholds of the MCC and GWAS score. In my opinion, a statistical test to select the HP combinations that perform statistically better over the 5 cross-validations should be done.

Lines 175-176 Results. The authors do not explain why 10 known AMR genes are sufficient to include the datasets with a minimum of 10 known AMR genes in the analysis. Also, the authors use a minority phenotype of > 5%. This could be an issue for the SVMs as explained below (see imbalance data).

Lines 192-197 Results. The authors mentioned that similar features were obtained when using the ensemble and Fisher's exact test. Have the authors tested the performance of the ensemble when using the features obtained with Fisher's exact test as input and compared it to the performance obtained with the pipeline they are proposing? How do the two pipelines compare?

Lines 434 -onwards the authors state that: " Briefly, protein sequences were clustered using CD-HIT v4.6 with minimum identity 80% and minimum alignment length 80%²⁴." And at line Line 494-onwards the authors state that: " For machine learning purposes, all features associated with a gene cluster containing a sequence linked to resistance for a drug by RGI or PATRIC text annotation were treated as known AMR features". The genes within the same cluster can vary up to 20% and such variation can include deletions, insertions, etc., which may cause a reduced/inactivated gene with respect to the linked resistance. However, all the clustered sequences are still treated equally as known functional AMR genes. Can the authors prove that such variability is not associated with reduced resistance phenotype and that those genes are truly underlying the targeted AMR profile? Can the authors justify why an 80% identity threshold is a good choice with respect to other homology values for clustering AMR features to be used for machine learning?

In line 527 Methods the authors state: "1) features present or missing in less than 3 genomes were removed", can the authors show the distribution of the features across the resistant/susceptible genomes considered, and how such distribution look after the classification? If a feature is associated with a specific resistance, how many genomes with that specific resistant phenotype (CLSI, etc.,) out of the total contain that feature? and likewise how many susceptible labelled genomes (CLSI, etc.,) for that specific antibiotic contain/do not contain that feature?

Line 528..” remaining features were sorted by log odds ratio (LOR) for resistant genomes, and features with the 25,000 highest and 25,000 lowest LORs were retained.” Did the authors test what happens when changing the threshold of the odds ratio? How does the performance of the ensemble change when selecting a lower threshold (less than 25.000)? And with this lower threshold are the features proportionally more connected to known AMR genes?

Lines 530-531 Methods. It is unclear how the selection of features and samples was done. Are the authors using replacement for both the selection of features and the selection of samples? On lines 530-531 the authors mention that they are using the BaggingEnsemble function from sklearn, however they do not provide the version of sklearn that they used. On the newest version 1.2.1 there is no BaggingEnsemble function only the BaggingClassifier function(<https://scikit-learn.org/stable/modules/generated/sklearn.ensemble.BaggingClassifier.html>). On the BaggingClassifier function only the samples are selected with replacement, the features are not (if the default values are used).

Lines 530-532. The authors as done in previous papers used a linear SVM. However, here they do not provide a reason for using a linear SVM. Some of the known disadvantages of using a linear SVM are 1. they do not perform well when the classes are overlapping (since the authors are using the intermediates, which could be a concern because the intermediates could be more closely related to susceptible samples) and 2. SVMs may present suboptimal results when dealing with imbalanced datasets (Table S4 indicates that 9 out of the 10 cases have imbalanced data).

Regarding the experimental validation of *cycA* and *frdD*, can the authors justify why these two were selected?

Response to Reviewers

Reviewer #1 (Remarks to the Author):

Global pathogenomic analysis identifies known and novel genetic antimicrobial resistance determinants in twelve species

Summary:

Hyun et al present a novel machine learning approach to identify resistance determinant by looking at a large number of public genomes with matched AMR phenotypes. They also describe the diversity and breadth of AMR genes within the dataset and across species/general class.

This manuscript was well-written with excellent figures.

Overall I was disappointed that there wasn't more effort to compare with existing bacterial GWAS studies and tools, and it was unclear if common problems in bacterial GWAS (like population structure) were addressed.

General comments:

- The authors mention AMR prediction but not existing GWAS approaches/studies in bacterial genomics in the introduction. This should be addressed.

Thank you for this valuable suggestion. We have expanded the introduction to mention existing tools for both human and microbial GWAS.

Introduction: ...Data-centric efforts to close this gap have focused on genome-wide association studies (GWAS), yielding tools for conducting the statistical tests underlying GWAS such as PLINK (Purcell et al. 2007) and GEMMA (Zhou and Stephens 2012), with some specializing in microbial GWAS by rigorously addressing population structure such as DBGWAS (Jaillard et al. 2018) and Pyseer (Lees et al. 2018). However, the predictive features identified when training ML models to predict AMR naturally provide a novel source of AMR gene candidates distinct from GWAS. Such analyses can leverage the extensive body of ML literature and tools to draw from a broader range of statistical and algorithmic frameworks than those previously used for GWAS, and as such...

- How much variation is there in the genomes the authors acquired from PATRIC? Are they both genetically diverse and well sampled (without specific bias from a single study, and with sufficient sensitive/resistant strains, for example?). This has implications on the conclusions for AMR sharing between species and predictive power for antimicrobial resistance.

Thank you for pointing out this issue regarding the scope of our findings. To examine the diversity of our dataset, we have visualized the distribution of genome MLST subtypes (from <https://github.com/tseemann/mlst>), source BioProject IDs (the best available metadata from PATRIC related to study), and number of susceptible/resistant genomes

for each species-drug pair (Figure S1) which is discussed in the text and in greater detail in the Supplemental Analysis. Broadly, we find that our genome collections are diverse from these three angles, and we note specific cases of lower coverage in the text.

Results: The overall diversity of the combined genomic and AMR data was evaluated by analyzing the distribution of MLST subtypes, genome BioProject IDs, and susceptible/resistant genomes by drug (Figure S1). Based on these evaluations, the genomes represented here are diverse with respect to subtype, study of origin, and resistance phenotypes, with cases of potentially lower coverage limited to species with AMR data for a smaller range of drugs (*N. gonorrhoeae*) or fewer genomes available in total (*C. coli*, *C. jejuni*) (see Supplemental Analysis).

Supplemental Analysis: The overall diversity of the genomes selected for this study was evaluated by examining the distribution of MLST subtypes (from <https://github.com/tseemann/mlst>) and BioProject IDs (from PATRIC) for each genome, and the number of susceptible/resistant genomes per drug (Figure S1). MLST distributions suggest that the genome collection for each species is genetically diverse, with the most common subtype per species never exceeding 50% of all genomes and at least 44 subtypes represented per species. BioProject ID distributions suggest that a majority of our genome collections also represent a wide range of studies, though the smallest collections were dominated by individual studies (*C. coli* and *C. jejuni* genomes originated almost entirely from PRJNA292668, *E. cloacae* genomes were majority from PRJEB5065) and genomes for *E. faecium* and *N. gonorrhoeae* had poor coverage regarding study of origin. Finally, genome counts by resistance suggest that both susceptible and resistant strains across a wide range of drug classes are present for most species, with only the *N. gonorrhoeae* collection potentially being limited due to AMR data being available for only two drugs.

Figure S1: Diversity of samples by subtype, study of origin, and AMR phenotype. a) Distribution of MLST subtypes and BioProject accession IDs for each species' genome collection. For MLSTs, the total number of additional MLST subtypes beyond the five most common are labeled. BioProjects comprising at least 50% of genomes for a given species are labeled. b) Distribution of AMR phenotypes by species, drug, and drug class. Drugs are sorted by drug class. The macrolide-lincosamide-streptogramin B drug class is abbreviated "MLSB".

- Generally the section on blaTEM in the results was a bit abrupt and did not seem to add to the overall aim of the manuscript, and as such could be relegated to supplementary. We have significantly shortened the discussion of blaTEM in the main text and transferred these results to the Supplemental Analysis.

- In the section starting on line 149, where the authors "...train models for both accuracy at predicting AMR phenotypes and biological relevance..." could the authors pick a few relevant examples where the model did well at predicting phenotype/known AMR genes? E.g. meropenem resistance based on blaKPC. At the moment I find it difficult to see solid examples.

Thank you for this suggestion to better communicate the results of our approach. We have added Figure S12 detailing the top 20 genetic features selected by our method for three species-drug cases: *Acinetobacter baumannii* vs. amikacin, *Escherichia coli* vs. ciprofloxacin, and *Klebsiella pneumoniae* vs. imipenem. Focusing on known AMR genes, these figures show how the SVM models identified 1) aminoglycoside modifying enzymes APH(3')-VI and AAC(6')-I as well as the 16S rRNA methyltransferase *armA* as predictive of amikacin resistance, 2) specific variants of *gyrA*, *parC*, and *parE* as well as plasmid-mediated quinolone resistance protein *qnrS* as predictive of ciprofloxacin resistance, and 3) beta-lactamases blaKPC, blaOXA, and blaNDM as predictive of imipenem resistance.

Figure S12: Visualization of the top 20 genetic features associated with AMR for three species-drug cases as identified by SVM ensembles. For each feature, the absolute value of the feature weight, overall rank, gene name, and feature type are shown. Features related to known AMR genes are colored and also labeled by whether they were also recovered by Pyseer and/or Fisher’s exact test, and other features are black. Features related to undercharacterized genes are labeled as either “hypothetical protein” or “mobile element”, numbered by the order the gene is displayed. Additional details are in Dataset S5.

- I generally found it a bit odd to compare the model to Fisher's exact test. Can the authors explain why they chose this? I wouldn't say it's commonly used, and there are many existing microbial GWAS tools available that could have been benchmarked against (Pyseer, bugwas, dbgwas).

Thank you for this excellent suggestion to better contextualize our method with existing microbial GWAS tools. We had initially opted to benchmark our method against Fisher's Exact test as one of the most common methods for testing for association between binary variables (i.e. the presence/absence of a genetic feature vs. susceptible/resistant phenotype) without requiring a separately calculated population structure. As recommended, we have added a similar benchmarking against Pyseer to make this section more specific to microbial GWAS (updated Figure 3, new Figure S13).

We find that Pyseer generally recovers slightly more known AMR genes than Fisher's exact test, but that overall the SVM ensemble approach still best recovers known AMR genes from the methods tested. Across the 127 species-drug cases tested, SVM ensembles recovered 263 known AMR gene-drug mappings, of which 123 were not recovered by either Pyseer or Fisher's exact test (compared to just 27 that were recovered by Pyseer or Fisher but not by SVM). We hope that these new results provide further evidence for the potential value of machine learning methods in tackling microbial GWAS.

Results: As a baseline level of AMR gene recovery, for each species-drug case, both Pyseer (Lees et al. 2018) and Fisher's exact tests were applied to estimate the strength of association between each genetic feature and the AMR phenotype, yielding population-adjusted and unadjusted p-values, respectively. Based on the number of known AMR genetic features recovered among the top 20 features (either by feature weight for SVM or p-value otherwise), the SVM ensemble approach broadly outperformed both Pyseer and Fisher's exact test. SVM ensembles identified more known AMR-associated features than Pyseer in 73 cases (57%), the same number in 38 cases (30%), and fewer features in just 16 cases (13%) (Figure 3c, Dataset S5). Nearly half of the equal performance cases (16/38, 42%) were instances where neither method could recover any known AMR features, and similar differences in performance were observed when comparing SVM ensembles and Fisher's exact tests (Figure S13a, Dataset S5). Across all 127 cases, SVM ensembles recovered 263 known AMR gene-drug mappings of which 123 were not recovered by either Pyseer or Fisher's exact tests, compared to just 27 mappings missed by SVM ensemble but recovered by Pyseer or Fisher's exact test (Figure 3d, Dataset S5). Similar proportions between the number of known AMR gene-drug mappings recovered per method were observed when examining the top 10 or top 50 features (Figure S13b-c). Examining SVM ensemble feature rankings, known AMR genes were distributed throughout the full range of ranks among the top 20 features per model, whereas those that were also recovered by other

methods were concentrated among the top 3 features (95/138, 69%) with nearly half being the top weighted feature of the corresponding SVM model (62/138, 45%) (Figure 3e). This result suggests that concordance between these three methods is mostly limited to features with the strongest statistical signals.

Figure S13: Additional comparisons between performance of SVM ensembles, Pyseer, and Fisher's exact test at recovering known AMR genes. a) Comparison between SVMs and Fisher's exact test at recovering known AMR genes across 127 species-drug cases. b-c) Total known AMR gene-drug mappings recovered by SVMs, Pyseer, and Fisher's exact tests across all cases, when defining a recovered gene as those in the top 10 or top 50 features when sorting by feature weight (for SVM) or p-value (for Pyseer and Fisher's exact test).

Specific comments:

- Line 74-75: Do the authors end up double-dipping certain variants if genes overlap and the authors are also taking 300 bp upstream/downstream? How does this affect the analysis?

Overlapping genes/genetic features were indeed a challenge in this study, making the interpretation of individual features potentially associated with AMR more difficult. For example, the challenge to interpretation is exemplified in our *frdD* case study, where our model identified a *frdD* coding variant as predictive of beta-lactam resistance, while we learned afterwards that the actual AMR mechanism was likely due to altering the promoter of beta-lactamase *ampC* which overlaps into the coding region of *frdD*. However, the more appropriate feature, the 300bp upstream variant of *ampC* with this promoter mutation, was neither correlated with the *frdD* coding variant nor as strongly associated with AMR, which is likely due to other mutations across the whole 300bp region reducing the signal from the specific causal *ampC/frdD* mutation. In general, an ORF-centric enumeration of genetic variation will inevitably result in many overlapping features (especially when attempting to account for flanking regulatory regions and/or complete transcriptional units), and any analyses of specific features should also examine overlapping features as we recommend in the Discussion.

For our specific results, we have designed the analyses to accommodate this issue. When comparing our SVM ensemble approach to Fisher's exact test and now Pyseer, all three methods are tested on the same set of genetic features and evaluated against the same set of known AMR genes, and the resulting comparison (Figure 3, Figure S13) includes how well each method tolerates overlapping/highly correlated features. When identifying candidate AMR-conferring genetic features, we filter out candidates that are highly correlated with known AMR genes (see Methods) which helps address instances when significant overlap with a known AMR gene causes an unrelated genetic feature to be associated with AMR.

- Line 138: Could the authors use blast (or something like BRIG) to determine from the 28 whole genomes which of these plasmids are more likely, rather than focusing only on the TEM contig? Are the genomes clonally related or is this more likely HGT?

Thank you for this suggestion, we have updated this section to discuss evidence for plasmid membership and HGT in these genomes more broadly. We have updated Figure S4b (previously Figure S3b) to now show all significant matches between PLSDb plasmids and the 28 genomes with TEM-116, determined using MASH as suggested by PLSDb as a faster alternative to blast-based methods. These results reflect how each of these genomes have evidence for exactly one of the three blaTEM-harboring plasmids observed. We have also updated Dataset S4 to include raw MASH outputs between the genomes and PLSDb.

Figure S4: Distribution of TEM-family beta-lactamases (blaTEMs) observed in 4,861 genomes across 8 species. a) Distribution of blaTEM alleles with respect to genome count, harboring species, and mutations relative to the TEM-1 allele. Alleles occurring in at least 10 genomes are labeled. TEM* and TEM** refer to unnamed blaTEM alleles. b) Alignments between genomes with TEM-116 and PLSDb plasmids based on MASH distance. Known plasmids carrying blaTEM are indicated. All plasmids with MASH distance < 0.025 to at least one genome with TEM-116 are shown. c) Distribution of the three plasmids containing TEM-116 by species. d) Relationship between beta-lactam associated AMR genes and cefoxitin minimum inhibitory concentration (MIC) in *S. aureus*. Numeric labels correspond to the number of genomes observed with the corresponding AMR genes and MIC.

To assess the relationship between these genomes, we examined the pairwise MASH distance matrices for *S. aureus* and *S. enterica* generated as inputs for Pyseer in inferring population structure, and compared pairwise distances between genomes with TEM-116 against pairwise distances between all genomes of those species (Figure S5). We find that the 17 *S. enterica* genomes with TEM-116 are highly similar and consistent with the source study confirming the genomes as derived from a single lineage, albeit from geographically diverse locations¹. However, as the plasmid distribution did not closely match phylogeny as inferred from MASH distances, it is ambiguous as to whether clonal dissemination or HGT played a larger role in forming the observed distribution of TEM-116 plasmids. In contrast, the 11 *S. aureus* genomes with TEM-116 were more genetically diverse, and the only *S. aureus* genome with the NZ_AJ437107.1 plasmid shared with *S. enterica* was genetically distinct from the other 10 genomes. Given the geographic prevalence of the TEM-116 *S. enterica* strains we believe that it is possible that *S. aureus* could have acquired blaTEM from *S. enterica* or another gram-negative strain. We have updated our Supplemental Analysis with this discussion.

Supplemental Analysis: Pairwise MASH distances between these genomes were compared to those between all *S. aureus* and *S. enterica* genomes to assess sequence similarity and clonality (Figure S5). The 17 *S. enterica* genomes with TEM-116 were highly similar with a median pairwise MASH distance of 0.0004 compared to a median of 0.0153 between all *S. enterica* genome pairs, consistent with the source study confirming all such genomes to be derived from a single clonal lineage of serotype Kentucky ST198, albeit from geographically diverse locations (Hawkey et al. 2019). However, as the presence of the two TEM-116 plasmids in these genomes do not closely track their phylogeny, it is ambiguous as to whether the observed plasmid distribution was established earlier in the lineage and resulted from clonal dissemination, or was a result of more recent horizontal gene transfer events.

In contrast, the 11 *S. aureus* strains with TEM-116 were genetically diverse with a median pairwise MASH distance of 0.0080 compared to a median of 0.0151 between all *S. aureus* genome pairs. The genome 1280.16776, which is predicted to carry plasmid NZ_AJ437107.1 shared by some *S. enterica* genomes, is genetically distinct from the other TEM-116 carrying *S. aureus* genomes. Given the geographic prevalence of the *S. enterica* genomes carrying NZ_AJ437107.1 (annotated as isolated across 10 countries: Djibouti, Egypt, France, Indonesia, Israel, Kenya, Kuwait, Morocco, Tanzania, Vietnam) (Hawkey et al. 2019), it is possible that the strain corresponding to 1280.16776 could have acquired this plasmid via horizontal gene transfer from a gram-negative species.

Figure S5: Sequence similarity between *S. enterica* and *S. aureus* genomes carrying TEM-116. a-b) Distribution of pairwise MASH distances between all 3,302 *S. enterica* and 2,248 *S. aureus* genomes. c-d) Clustermap based on pairwise MASH distances between the 17 *S. enterica* and 11 *S. aureus* genomes carrying TEM-116. Heatmaps share the same color scales. Genomes are colored by their predicted blaTEM plasmid and are clustered using single linkage and Euclidean distances.

- Line 146: I think it's worth mentioning that the authors have only n=4 samples here with all three genes. Are these independent genomes or is there an unobserved lineage effect?

Thank you for this comment, we have updated the section (now in the Supplemental Analysis) to emphasize the small sample size, and examined the diversity of those four strains. All four strains originate from a single study (BioProject: PRJNA433074) on a pork production chain in China but span two host types (human, pig) and two MLST subtypes (ST9, ST59) making it unlikely that this is an unobserved lineage effect.

Supplemental Analysis: While only four genomes with blaTEM also had MIC data, those genomes (which also carried mecA and blaZ) had significantly elevated MICs compared

to those with just *mecA* and *blaZ* or *mecA* alone ($p=0.0074$, Mann-Whitney U-test), suggesting blaTEM may confer additional protection from beta-lactams in already resistant *S. aureus* (Figure S4c, Figure S6). These four strains were isolated from a study of the pork production chain in Shandong, China (BioProject: PRJNA433074) (Bi et al. 2018), reflecting a geographically confined but genetically diverse sample set. The strains span two MLST subtypes (ST9, ST59), with three isolated from workers (PATRIC IDs: 1280.16838, 1280.16865, 1280.16862) and one from a pig (1280.16776).

We have also qualified the mention of this result in the Discussion to say that blaTEM was observed in the most resistant *S. aureus* strains rather than increasing resistance, given the weak evidence due to small sample size.

A case study of blaTEM revealed one variant, TEM-116, to be present in the GP species *S. aureus* and found among *S. aureus* strains with the highest levels of resistance to beta-lactams.

- Figure 2d: why are there three columns that all say blaTEM cluster? different TEM alleles?

Thank you for pointing out this ambiguity. We have updated the figure legend to mention that these are three different sequence clusters (found by CD-HIT) all related to blaTEM, with the first representing full variants and the other two representing fragments.

The three “blaTEM cluster” columns correspond to different sequence clusters identified by CD-HIT all related to TEM family beta-lactamases, with the first representing full variants and the others representing fragments.

- Line 163: "A feature was labeled as a known AMR genetic feature if its corresponding gene cluster was a known AMR gene for the drug of interest as annotated by RGI and PATRIC" - what about SNPs conferring resistance? e.g. *rpoB/gyrA* are not resistance genes, but SNPs in them can confer resistance.

Thank you for pointing out this potential issue. Indeed, SNPs are not directly modeled in our feature set, but are captured by the variant-level features which contain various mutations relative to the wildtype variant of the corresponding gene (usually defined as the most commonly observed variant). At the onset of this analysis it was not feasible to manually annotate all individual mutations for SNP-mediated resistance across all species and drugs. Thus, for model validation purposes, we assumed that within AMR genes (i.e. those with known SNP-mediated resistance), the presence of one variant over another is likely to impact resistance, and labeled all such features as known AMR features.

To examine this assumption, we have added a new analysis of all instances of our

models associating a *gyrA* allele with fluoroquinolone resistance (30 alleles across 7 species). All alleles positively associated with resistance contained at least one known resistance-conferring mutation, and those negatively associated contained no such mutations (see Supplemental Analysis and Dataset S5). Given these results, we believe that our approach of labeling known AMR genes at the allele-level is capable of capturing SNP-mediated resistance, and is also unlikely to overestimate the capabilities of the SVM at recovering known AMR genes.

Results: Finally, a detailed analysis of the 30 *gyrA* alleles identified by SVM as associated with fluoroquinolone resistance finds all such variants to be consistent with the current literature on *gyrA*-mediated resistance; all positively-associated alleles carried at least one known resistance-conferring substitution, while all negatively-associated alleles carried no such mutations (Supplemental Analysis, Dataset S5).

Supplemental Analysis: All *gyrA* alleles among the top 50 features from models related to fluoroquinolone resistance were identified, totaling 30 alleles across 7 species. Mutations were called relative to the wildtype allele of the corresponding species, defined as the most commonly observed *gyrA* allele among genomes susceptible to at least one fluoroquinolone. Resulting mutations and GenBank accession IDs for wildtype alleles are available in Dataset S5. All recovered *gyrA* alleles with positive feature weight for resistance had at least one known resistance-conferring mutation, covering the following substitutions: S81L in *A. baumannii*, T86I in *C. coli* and *C. jejuni*, S83L and D87N in *E. coli*, T83I in *P. aeruginosa*, and S83Y and D87* in *S. enterica*. All recovered *gyrA* alleles with negative feature weight for resistance had no such known resistance-conferring mutations. These results suggest that the *gyrA* alleles recovered by the SVM ensemble approach are consistent with current understanding of the *gyrA* mutational landscape with respect to fluoroquinolone resistance.

- Line 166: I am unclear on whether the authors trained and evaluated the model on the same data, or on different subsets. Could the authors explain this more, and their rationale?

All of our model evaluations were conducted through 5-fold cross validation experiments. Cross validation generates multiple pairs of non-overlapping training/testing sets from the full dataset and is a popular method for assessing the generalizability and robustness of machine learning models². Briefly, for a given species-drug case, the set of relevant genomes + phenotypes is randomly split into five evenly sized groups, and then for each of five “folds”, a model is trained on four groups (training set) and evaluated on the 5th (test set), with the test group rotating for each fold. In Figure 3b, we report the average accuracy (using Matthews correlation coefficient) on the test set and average GWAS score across the five folds.

- Line 182: "accuracy did not guarantee biological relevance" Is this because of strong population structure? Does the model account for this at all? This is a well known problem with bacterial GWAS

It is difficult to link this observation to any single reason, but we suspect that having many highly correlated features, usually stemming from population structure, plays a significant role. While population structure is not addressed directly by our model, we have previously shown that randomly subsampling both the feature and sample sets during training (which can help break apart such correlations) appears to improve recovery of known AMR genes without compromising AMR prediction accuracy³. We had also attempted to address population structure directly in that study by balancing the representation of subtypes in the training data through over/undersampling, but did not observe any consistent improvement in either accuracy or known AMR gene recovery. We have added this issue to the Discussion.

Discussion: This is likely due to strong population structure resulting from the clonal nature of bacteria, which can lead to significant correlations between causal and hitchhiker mutations that are difficult to distinguish using statistical approaches (Power et al. 2017). Though the SVM approach presented here does not directly address population structure, it was previously shown that attempting to do so through weighted sampling of strain subtypes did not improve AMR gene recovery over random sampling (Hyun et al. 2020). While other tools with different approaches for addressing population stratification may outperform those tested here, ML approaches can supplement AMR gene recovery without requiring the often computationally expensive task of defining population structure in advance.

- Line 183: "Certain dataset parameters were weakly but significantly associated..." I don't see this in the S6c plot, what is significant here?

The associations discussed here are based on the tests in Table S8 (previously Table S7), which show statistically significant associations between those parameters and model performance. However, the effect sizes are weak and not visually apparent, as you correctly point out in the corresponding plot Figure S6c (now part of Figure S10a).

- Line 190: "The top 50 features for each model are available in Dataset S5, and their sequences in Dataset S6." I may have misunderstood, but the authors have >100 models, so should the lists in these datasets be much longer?

This is correct, the sheet "Model_Top50s" in Dataset S5 has 6,350 entries, corresponding to 127 models x 50 hits. Dataset S6 also has on the order of 1000s of sequences (not exactly 6,350, as some features may be in the top 50 for multiple models, such as models for the same species across drugs from the same class).

- Line 256: Were there any baseline growth differences between WT and delta-cycA? What do the results look like with only CIP in rich/minimal media, without supplementing substrates?

Among the 120 conditions tested, only the six labeled in Figure 5b resulted in statistically significant differences in final density between the WT and *cycA* KO strains (all of which involved M9 media), suggesting that the loss of *cycA* does not significantly impact growth in rich media and rarely in minimal media. These 120 conditions include those without any substrate supplementation (referred to as supplement = None) with the original measurements in Dataset S8. A majority of the significant cases involve M9 + D-serine, which became the focus of the *cycA* analysis.

- Line 261: it appears the *cycA* knockout is doing comparably/better to the WT - the authors haven't explicitly said this in the results here. What does this mean for the authors ML model, since it predicted the WT associated with quinolone resistance?

Thank you for pointing out this issue. We believe that this seemingly conflicting result between the experimental validation and statistical analysis (both from our ML model and simpler statistical associations in Figure 5a) highlights the importance of media on resistance. As most publicly-available AMR measurements for *E. coli* used in this study were conducted in rich Mueller-Hinton media or its variations (such as CA-MHB), observing the *cycA* knockout to perform better only in M9 minimal media does not directly conflict with the statistical analysis. However, given that no significant difference between WT and KO was observed in CA-MHB suggests that the precise prediction of the WT being associated with resistance in rich media may have been a false positive, while the more general prediction of *cycA* being associated with resistance appears to be correct if possibly coincidental.

- Line 297: did the authors sequence the strain after the experiment? Is it possible it acquired the mutation?

Thank you for this comment. Though we did not save these specific strains and are consequently unable to sequence them, we agree that acquiring the single *frdD/ampC* point mutation to convert these strains to possess the known *ampC* overexpressing promoter is a plausible explanation for the delayed growth, and have updated the Results to mention this theory.

*Results: ...and may have resulted from acquiring the single point mutation necessary to yield the known *ampC* overpressing promoter.*

- Line 426: "For each species, genomes were filtered to those meeting the four criteria previously described" please add the specifics for reproducibility

We have updated the Methods to describe the four criteria for genome selection from our previous work.

Methods: ... 1) genome status is "WGS" or "Complete", 2) number of contigs is within 2.5 times the median number of contigs across all assemblies for that species, 3) number of annotated CDSs is within 3 standard deviations of the mean, and 4) total genome length is within 3 standard deviations of the mean.

- line 427: What is "fine consistency"? I haven't heard this term before
Fine consistency is a metric provided by PATRIC for evaluating the quality of a genome's functional annotations, implemented by the EvalCon tool⁴. This was previously described incorrectly as being part of CheckM, and we have updated the Methods to correct this error.

- Line 437: How did the authors obtain 300bp upstream/downstream, and what happened if there was a contig break before this?

We have updated our Methods to make the identification of ORF-flanking sequences more clear. The flanking 300bp variants of a gene were identified by locating all occurrences of all alleles of the gene in their original genome assemblies and extracting the 300bp directly flanking those occurrences. Cases where the 300bp region would be interrupted by a contig break were ignored, as instantiating a new shortened variant for such a case would yield a feature unique to the specific genome that would be uninformative for machine learning purposes (the corresponding genome would instead be treated as having none of the identified 5' or 3' variants).

Methods: 5'/3' variants of a gene were identified by locating all occurrences of all alleles across all genome assemblies and extracting the 300bp directly flanking those occurrences. Cases in which a 300bp flanking region is interrupted by a contig break were ignored (for ML purposes, such genomes were treated as not having any specific 5'/3' variant).

- Minor:
 - line 17: specify "multiple phylogenetic classes"
 - line 125: remove "which"
 - Line 162: remove "a"

Thank you for catching these errors, we have updated the text as suggested.

Reviewer #2 (Remarks to the Author):

Hyun and colleagues present a study on AMR. In the paper, they track AMR genes across diverse genomes, build SVM models to identify features that are associated with AMR in these genomes, and experimentally verify that two mutations – a deletion in *cycA* and a V111D variant in *frdD* – have a role in resistance. Overall, I think the paper has novelty and is a nice contribution to the field. The science appears to be sound and most of my comments relate to clarifying the text.

- Line 18 what do the authors mean by the traditional methods?
Thank you for bringing attention to this wording. We have updated the Abstract and main text to instead describe commonly used GWAS methods such as those our workflow was compared against (Pyseer and Fisher's Exact test) as "contemporary" rather than "traditional", to reflect their current active usage in GWAS.
- Line 21 The reference genome for the variant V111D needs to be declared here and elsewhere in the paper. I was originally thinking this was in MG1655, but it turned out not to be. Also, it is disorienting because the previous sentences are talking about resistance across the tree of life, and then this zooms in on an *E. coli* mutant, which isn't bad, it just needs clarification. Is BW25113 always your reference strain?
Thank you for pointing this out, we have updated the abstract to specify that the candidates were validated in the context of *E. coli* BW25113. While our usual reference strain is MG1655, we opted to use BW25113 for these validations to make use of the Keio knockout collection, which used BW25113 as its base strain. With regards to the candidates, we confirm that the BW25113 and MG1655 reference genomes (accession IDs NZ_CP009273.1 and U00096.3, respectively) are identical across all positions within 40kb of *frdD* or *cycA*. We have added this reasoning and finding to the Results.

Results: E. coli BW25113 was chosen as the base strain for validation to make use of the Keio knockout collection (Baba et al. 2006), which is genetically identical to K12 MG1655 across all positions within 40kb of *frdD* and *cycA* based on reference genomes NZ_CP009273.1 and U00096.3. Wildtype (WT) *cycA* and *frdD* were defined as the most common allele of the respective genes observed across all *E. coli* genomes in this study, which were also the alleles present in the BW25113 and K-12 MG1655 reference genomes.

- Based on Figure 1B, it looks like *M. tuberculosis* was excluded from the analysis even though it might have the most AST data. Was there a biological or strategic reason for this? It has a lot of SNP-based AMR, but also perhaps a lot to learn about.
Thank you for bringing up this issue. A similar analysis of *M. tuberculosis* would certainly be a valuable application and test of our method, and its exclusion was incidental rather than strategic. The choice of species was taken directly from our previous work⁵, which

selected species among the ESKAPEE pathogens and the WHO priority pathogens list and did not include *M. tuberculosis*⁶. We have updated the Methods to explain our choice of species.

Results: ...by taxon ID to 12 species among the ESKAPEE pathogens or WHO global priority pathogens released in 2017.

- Figure 1D MLSB should be defined in the text or figure legend
Figure legends have been updated to define MLSB as the drug class “macrolide-lincosamide-streptogramin B”.
- I think the authors need to be careful to fully qualify any statements comparing their SVM approach to GWAS. On line 362 in the discussion, they describe some of the more sophisticated tests for population stratification etc. While I do agree that doing this type of analysis would be inconvenient, the authors do not compare their approach with these more rigorous methods. That makes it difficult to say that it is better or outperforms GWAS.

Thank you for pointing out this issue with the analysis. Following similar suggestions from another reviewer, we have added a direct comparison between our approach and Pyseer (Figure 3, Figure S13, Dataset S5), a more modern tool for microbial GWAS that addresses population structure directly. While this comparison was still favorable for our SVM approach, we have been careful to qualify the merits of ML for GWAS purposes in the Discussion and focus on other benefits of the approach (i.e. improving AMR gene recovery without requiring the often computationally expensive task of calculating population structure in advance, as required for many existing microbial GWAS tools).

- It looks like some of the SVMs are based on the presence/absence of genes, is that also true for the GWAS comparison?
Yes, all of our SVM models were trained using presence/absence calls of genes and their various associated features (individual alleles, 5'/3' variants), and AMR phenotypes. To leverage a trained model for GWAS purposes, we sort all features by the weight assigned by the model, and examine the highest weighted features such as the top 10/20/50 features for known AMR genes and later, for potential AMR gene candidates (details in Figure 3a and the Methods).

For comparison against existing GWAS tools, Fisher's exact test and Pyseer were provided the same set of feature presence/absence calls and AMR phenotypes, both of which calculate a p-value for each feature. Those features are sorted by p-value and the top 10/20/50 features are examined for known AMR genes, to enable the comparison between these tools and our SVM approach as in Figure 3c-e and Figure S13.

- The authors ultimately ended up talking about amino acid level changes (particularly in the V111D mutant). But it was not clear to me how the upstream sequences were turned into features that allowed this to be detected by the SVM model. Were these detected after the fact?

We have expanded the 2nd section of the Methods to include greater detail on how flanking upstream/downstream features were identified. Briefly, once a gene and its coding sequence variants or “alleles” have been identified (from ORF clustering), all instances of those alleles are identified in the original genome assemblies, and the 300bp flanking all such instances are extracted to create a set of 300bp sequences that comprise all observed upstream/downstream variants of that gene. The presence/absence of each of these variants are then provided as features for training SVMs, alongside previously mentioned features (genes, alleles).

Methods: 5'/3' variants of a gene were identified by locating all occurrences of all alleles across all genome assemblies and extracting the 300bp directly flanking those occurrences. Cases in which a 300bp flanking region is interrupted by a contig break were ignored (for ML purposes, such genomes were treated as not having any specific 5'/3' variant).

Upstream sequences identified as AMR gene candidates have been interpreted relative to their corresponding “wildtype” upstream sequence, which are defined as the most commonly observed 300bp sequence flanking the corresponding gene (analogous to how “wildtype” was defined for coding sequences, i.e. for *frdD* and *cycA*). These interpretations are available in Dataset S7 and the relevant sequences in Dataset S6.

- Is *frdD* plasmid-borne? Do *frdD* and *ampC* always travel as a pair? If not, are there equivalent promotor mutations in upstream genes where *ampC* is located elsewhere that were found in the analysis?

Thank you for these interesting questions. We find that both *frdD* and *ampC* are core genes in *E. coli* (found in 98.4% and 97.9% of all *E. coli* genomes, respectively), which suggests that they are more likely chromosomal than plasmid-borne. Regarding co-localization, we find the distance between *frdD* and *ampC* to be extremely consistent across *E. coli* genomes, with 3,698 out of 3,748 genomes (98.7%) with both *frdD* and *ampC* to harbor both genes on the same contig with ORFs separated by exactly 63bp. We have added these new results to the Supplemental Analysis, which are referenced in the *frdD* section of the Results.

Results: This proximity was confirmed globally, with 3,698 (96%) *E. coli* genomes harboring both *frdD* and *ampC* on the same contig with ORFs spaced by exactly 63bp (see Supplemental Analysis).

Supplemental Analysis: Analysis of all 3,856 *E. coli* genomes in this study suggests that *cycA*, *frdD*, and *ampC* are core genes of *E. coli*, found in 3,823 (99.1%), 3,796 (98.4%), and 3,775 (97.8%) of genomes, respectively. No genomes had multiple copies of *cycA*, *frdD*, or *ampC*. 3,748 (97.2%) genomes had both *frdD* and *ampC*, of which all but three harbored *frdD* and *ampC* on the same contig. The distance between the *frdD* and *ampC* ORFs was highly consistent with a mean of 62.9bp, standard deviation of 5.2bp, and range of 30-192bp. A vast majority of genomes (3,698) had a *frdD-ampC* distance of exactly 63bp. This result suggests that a *frdD* V111D mutation will impact the *ampC* promoter in most *E. coli* strains. All instances of *cycA*, *frdD*, and *ampC* identified across all *E. coli* genomes are available in Dataset S8.

- I felt that nearly all of the description of the hyperparameter selection could be moved to the methods. It really slowed down the results section.
The discussion of HP optimization was significantly shortened in the main text and the analysis was moved to the Supplemental Analysis.
- Line 91: What do they mean by best annotation?
Thank you for pointing out this ambiguity, we have clarified the text to say “...with the greatest number of AMR genes identified for major drug classes...”
- Line 243: I got hung up on the term “coding variant” and how that relates to the deletion mutant that is ultimately reported.
We have updated the text to use the term “allele” consistently when referring to individual variants of a gene’s translated protein sequence (i.e. variants involving the gene’s coding region as opposed to flanking noncoding regions), and have expanded the description of the two candidates tested relative to *E. coli* reference genomes.
- Line 244 what species does the *cycA* refer to, and what is wild type
We have updated this section to mention that the validations were done in the context of *E. coli*, and that wildtype variants (for both *cycA* and *frdD*) were defined as the most commonly observed variant across all *E. coli* genomes analyzed in this study. These wildtype variants are identical to those in the K12-MG1655 and BW25113 reference genomes.

Results: *E. coli* BW25113 was chosen as the base strain for validation to make use of the Keio knockout collection (Baba et al. 2006), which is genetically identical to K12 MG1655 across all positions within 40kb of *frdD* and *cycA* based on reference genomes NZ_CP009273.1 and U00096.3. Wildtype (WT) *cycA* and *frdD* were defined as the most common allele of the respective genes observed across all *E. coli* genomes in this study, which were also the alleles present in the BW25113 and K-12 MG1655 reference genomes.

- Why does figure 5C go: 8, 16, 31, 62, 125 instead of 8, 16, 32, 64, 128? Also on 5C, why isn't 31 significant? It seems odd that 16 ug/L has the greatest difference and then the trend almost disappears

Thank you for pointing this out, we have corrected Figure 5b to have consistent concentrations with Figure 5c. For the choice of concentrations, we opted to test concentrations that matched MICs commonly reported in AMR data from PATRIC (Dataset S2). We believe this trend results from many such measurements following two-fold serial dilutions starting at 1mg/L, leading to concentrations 500, 250, 125, 62, 31, 16, 8ug/L.

For the 31ug/L case, the original density measurements were slightly noisier than of the other cases, and thus the result is not statistically significant after multiple hypothesis correction. For the 16ug/L case, we agree it is unusual that it presents the largest difference between KO and WT and deviates from the increasing KO/WT trend with respect to CIP concentration. While outside the scope of the current study, one potential explanation that may be worth future investigation is that there is a synergistic antimicrobial effect between D-serine and sublethal concentrations of CIP that is prevented by the loss of *cycA*, hence resulting in the much larger decrease in cell density between 8 and 16ug/L for WT than KO.

- Line 32 Reference should be updated to <https://pubmed.ncbi.nlm.nih.gov/34379107/>
Thank you for catching this, we have replaced the reference.
- Line 427 In the BV-BRC resource, fine consistency in genome quality comes from this paper, not checkM <https://pubmed.ncbi.nlm.nih.gov/31581946/>, although CheckM is often computed as well.
Thank you for correcting this error, we have updated the Methods appropriately and have cited this publication.
- I was hoping for some discussion of how the authors ultimately decided to validate the *cycA* and *frdD* mutants, when there were hundreds to look at. Were there some that you couldn't work out? Based on my reading of this there were other features that ranked higher.
Thank you for indicating the lack of detail in this process. We have added a new section to the Supplemental Analysis ("Selection of AMR gene candidates for experimental validation") that describes additional decisions not previously mentioned in the Methods as well as intermediate results throughout the process of translating the top features from our ML models, to the 142 proposed candidates, and to the final choice of *frdD* and *cycA* for validation. In brief, the choice was based on an examination of functional annotations and cross-drug evidence not included in the ML training process, as well as

our own limitations in the types of features we were best equipped to test experimentally.

In more detail, the initial reduction from the top model features to the 142 candidates follows Figure 4a and is detailed in Dataset S7. From top model features, those associated with known AMR genes or observed very rarely were removed first, and remaining features were scored based on 1) the number of drugs from the same drug class with which they were significantly associated with resistance, and 2) the number of drugs for which there exist resistant genomes without any known AMR genes that could be explained by the feature. The top 10 features by this score for each (species, drug class) pairs yielded the 142 candidates.

These were further filtered to exclude features that were poorly annotated, related to mobile elements, or associated with AMR for unrelated drugs to yield a smaller set of 43 “well-characterized” candidates (also detailed in Dataset S7). Just six of these referred to specific sequence variants in *E. coli* for which we were best equipped to validate experimentally. We ultimately decided to focus on coding variants to take advantage of the Keio knockout collection which left us with four options, from which we chose *frdD* and *cycA* due to their better functional and metabolic characterization compared to *sugE* and *yjfN*.

- Line 262: I found this sentence to be awkward. “Across all conditions involving D-serine, while increasing CIP concentration reduced final density for both strains, the KO strain achieved higher densities for 16-125µg/L CIP, but only in M9 media and not in CA-MHB (Figure 5c)”.

We have re-worded the sentence as follows: “In conditions involving D-serine, while increasing CIP concentration reduced final density for both strains, the KO strain was conditionally less sensitive. The KO strain achieved higher densities than WT for 16-125µg/L CIP, but only in M9 media and not in CA-MHB (Figure 5c).”

- 272: “response that is adapted to neither stress” I had difficulty understanding this sentence.

We have updated this phrase to say “response that is neither optimized for D-serine nor CIP”

- I found the *CycA* section in the results to be generally difficult to read, but I think that the authors are suggesting a synergistic killing effect between cipro and D-Serine in M9 media (when L-serine can’t compete with D-serine), which is fine, but what conditions in nature would ever cause this? Why would this ever evolve? Is there some other real-world compound that is transported by *CycA* that is the actual physiological culprit for the evolution of this conditional resistance mutant? Is there a scenario where D-serine is made, like in a macrophage?

Thank you for the careful reading into the *cycA* results. We agree that such a conditional

mutant would require highly specific conditions to be selected for and is likely very rare (>99% of the *E. coli* genomes we examined carried some variant of *cycA*). However, previous studies have identified several (human) host regions with significant amounts of D-serine and with evidence that the D-serine is of endogenous origin, such as in urine where concentrations from 0.1-1mM were commonly observed with implications for UPEC⁷⁻⁸. In such environments where the host is also receiving antibiotics, it may be possible for a *cycA* loss-of-function mutant to emerge, if the synergistic killing effect between CIP and D-serine is real.

Reviewer #3 (Remarks to the Author):

The work presented by Hyun et al. describes a workflow that combines pangenomics and machine learning (ML) to identify known and novel AMR genes. Using data retrieved from public repositories and encompassing 12 species spanning 27,155 genomes and 69 drugs. By applying this pipeline, they found that AMR gene transfer is mostly confined to related species. In this study, they identified 142 novel AMR gene candidates, two of which were studied experimentally and showed conditional resistance.

A few considerations and concerns are reported below:

- Can the authors discuss what are the improvements (e.g. performance, genes found) obtained by the optimization of the hyperparameters, as done in this work, in comparison to the results found in Hyun et al. PLoS Comput. Biol. 16, e1007608 (2020)?

In terms of raw performance, we find that the addition of hyperparameter optimization to our previous work offered consistent but only minor improvements to both phenotype prediction accuracy and recovery of known AMR genes (Figure S7, comparing “default” hyperparameters from the previous work vs. optimized). However, one practical improvement uncovered through this process was that the size of the ensemble rarely needed to exceed 50 estimators to maximize performance (compared to the previous work using 500), and that smaller, more computationally efficient ensembles can be used in future iterations without compromising performance. These findings are discussed in the 4th paragraph of the Discussion.

- Line 15-17 abstract: “found AMR gene transfer mostly confined within related species” see also Line 95 Results and Line 330 onwards and discussion. The authors claim that they found AMR gene transfer, mentioning that their results were consistent with previous studies [REF 43 and 44]. I respectfully disagree with this assertion and with the claim that the authors can provide evidence for gene transfer in their work. In Ref 44 a collection of 2173 bacterial isolates from healthcare-associated infections from a single hospital over 18 months was performed. In that work, the authors performed a collection campaign where samples were collected in a specific environment and time following an appropriate protocol to track the appearance of plasmid transmission and to provide evidence of plasmid transfer. The work presented by Hyun et al. would require additional factors to translate their findings into transfer and to make inferences on gene transfer. Co-presence of AMR features does not imply transfer.

Thank you for this detailed assessment regarding our analysis of multi-species AMR genes. We agree that the analyses we conducted are insufficient for positive confirmation of gene transfer, and we have qualified our conclusions regarding the blaTEM case study (now moved to the Supplemental Analysis) to reflect this. However, we believe that the extreme rarity of cross-species AMR genes that we observed in this analysis (specifically, those that cross phylogenetic class boundaries) provides negative

confirmation on AMR gene transfer, i.e. co-presence doesn't imply transfer, but lack of co-presence is strong evidence against against transfer. As the claim in our abstract currently concerns the limits of AMR gene transfer and depends only on negative confirmation, we have opted to keep the claim in the Abstract as is.

- Line 155-157 Results and 524-526 Methods. The ML framework proposed by the authors classifies genomes into susceptible vs non-susceptible. To do this the authors group the isolates with intermediate phenotypes with those with resistant phenotypes. However, considering that intermediate values can range from a minimum value that is close to the max. value of susceptibility to an upper value that is close to the minimum value of resistance, including the intermediate cases in the resistant group can create a bias (which could be a concern because the intermediates could be more closely related to susceptible samples). For example, Kim et al [REF 13] and others, when using machine learning and CLSI breakpoints to determine the antibiotic phenotype (sensitive versus resistant) from sequencing data, did not include the isolates with intermediate susceptibility in the analysis. Mislabeling observations in classification tasks can strongly affect the true outcome obtained by the learners.

Thank you for this insightful comment on processing AMR phenotypes. We have expanded our previous analysis of dataset parameters to examine whether the fraction of genomes originally annotated as “intermediate” was associated with model performance across our 127 species-drug cases (Table S8, Figure S10). Indeed, an increasing fraction of intermediate genomes was mildly negatively correlated with model performance (Spearman correlation = -0.359 with test set MCC in 5-fold CV, -0.103 with number of AMR genes recovered), with the association with MCC being statistically significant. However, we do not believe that reprocessing our AMR phenotypes would significantly impact our final predicted AMR gene candidates. Among the 87 “accurate” models (those with MCC > 0.8) from which we selected our candidates, the median fraction of genomes used for training with the “intermediate” phenotype was only 0.8%. We have added these results to the Supplemental Analysis, and have provided all the original SIR and MIC measurements for all (genome, drug) pairs in Dataset S2 for future analyses targeting more granular predictions of AMR phenotype.

Supplemental Analysis: The significant result regarding intermediate genomes points to an opportunity to improve the overall workflow. Given the negative association between intermediate genomes and model performance (likely due to their genetic similarity to both susceptible and resistant genomes), future iterations of this workflow may benefit from treating intermediate genomes as a separate phenotype, and original SIR and MIC phenotypes are available in Dataset S2. Nonetheless, the prevalence of intermediate genomes and their potential confounding effects is unlikely to have a significant impact on the final set of AMR gene candidates, as the accurate models (those with MCC > 0.8) used for identifying candidates had a median intermediate genome fraction of only 0.8%.

Table S8: Impact of input data properties on SVM ensemble performance. Impact was assessed with Kruskal-Wallis tests for categorical properties (species, drug class) and Spearman correlation for quantitative properties (number of genomes, minority phenotype fraction, **fraction of genomes with the “intermediate” phenotype, total number of known AMR genes**). Rows are sorted by p-value. **Cases significant to FWER < 0.05 (Bonferroni correction, 12 tests) are starred.**

Performance Metric	Data Metric	Statistical Test	p-value
AMR genes in top 20	num. genomes	Spearman-R	<0.00001*
Test MCC in 5CV	species	Kruskal-Wallis	<0.00001*
Test MCC in 5CV	intermediate fraction	Spearman-R	0.00003*
AMR genes in top 20	total known AMR genes	Spearman-R	0.00079*
AMR genes in top 20	minority fraction	Spearman-R	0.00464
Test MCC in 5CV	num. genomes	Spearman-R	0.00537
AMR genes in top 20	species	Kruskal-Wallis	0.00721
Test MCC in 5CV	total known AMR genes	Spearman-R	0.01536
AMR genes in top 20	intermediate fraction	Spearman-R	0.25039
Test MCC in 5CV	drug class	Kruskal-Wallis	0.30250
AMR genes in top 20	drug class	Kruskal-Wallis	0.48732
Test MCC in 5CV	minority fraction	Spearman-R	0.95993

Figure S10: Relationship between dataset parameters and model performance across 127 species-drug cases. a) Performance of hyperparameter optimized SVM ensembles vs. dataset size, extent of class imbalance, abundance of “intermediate” resistant genomes, and total known AMR genes annotated. Spearman correlation coefficients are shown. d) Performance of the 127 SVM ensembles versus drug class.

- Line 159-163 Results and Line 533-538 Methods. The authors propose a “GWAS score” as a metric to define the biological relevance of the AMR genes. Is this score normally used in GWAS studies? If yes, can the authors provide references; if not how the authors defined this score? Was it defined empirically? What is the reasoning behind it and the robustness to support its use? I could not find references in support of the robustness of this score and more importantly no benchmarking of the method.

The “GWAS score” presented here was inspired by some of the informal intuition behind how follow-up analyses are often conducted for the top hits from a GWAS study. When hundreds of significant hits are observed, we find that many studies (both GWAS and ML) will focus their attention towards the top X features by effect size (i.e. top 3, 5, 10, 20). Rather than picking one of these cutoffs, we mimic the diminishing attention with respect to rank by giving exponentially diminishing marginal GWAS score per recovered known gene as rank increases (specifically, to diminish by half every 10 ranks).

This rationale makes this score difficult to formally benchmark, and it is unclear what the appropriate true value for AMR gene “known-ness” would be for this function (effect size, annotation completeness scores, etc.). As much of the final results are reported in the simpler context of “known AMR genes in the top 20 hits”, we have opted to de-prioritize the GWAS score in the Discussion. We have added this rationale to the Supplemental Analysis.

Supplemental Analysis: The “GWAS score” was inspired by knowledge of follow-up analyses from past association studies. When hundreds of significant hits are observed, many GWAS and ML studies of AMR will focus on the top X features by effect size (i.e. top 3, 5, 10, 20) (Kim et al. 2020), (Nguyen et al. 2019), (Earle et al. 2016). The GWAS score mimics this diminishing attention with respect to rank by giving exponentially diminishing marginal GWAS score per recovered known gene as rank increases, specifically, to diminish by half every 10 ranks.

- The authors mention that with their method they can identify novel genes but if the GWAS score is defined as: “.a GWAS score based on the rankings of known AMR genes among a model’s predictive features” how can such a metric, which seems to strongly rely on what is known, correctly identify novel AMR genes (which can differ from the known ones)? The method seems biased towards the known AMR genes and might not fully address the novel genes.

Thank you for this insightful comment. This is indeed a fundamental limitation of the current approach, where its generalizability relies on the assumption that improved recovery of known AMR genes can translate to improved recovery of novel AMR genes.

While this assumption is impossible to validate definitively, we conducted an assessment

of the generalizability of the GWAS score by randomly hiding half of our known AMR genes and seeing if GWAS scores computed using the visible genes are correlated with the GWAS scores computed using hidden genes. In detail, for each of our 10 test species-drug cases (previously referred to as “representative” cases), we 1) randomly hide half of the known AMR genes, 2) compute GWAS scores using either visible or hidden AMR genes across all 256 tested hyperparameter combinations, 3) compute the Spearman correlation between these two GWAS scores, and 4) repeat steps 1-3 100 times to compute a distribution of correlation coefficients (Dataset S5).

In general, we find that the GWAS score generalizes well to hidden AMR genes, with Spearman correlations between GWAS scores from visible vs. hidden AMR genes greater than 0.4 for 6/10 cases, and greater than 0.6 for 4/10 cases (Figure S9). We also find that the capacity for the GWAS score to generalize is strongly dependent on the initial level of variation in GWAS scores between hyperparameter combinations (Figure S9b), i.e. if all models have highly similar GWAS scores, the ranking of models by GWAS score is not robust and results in weak correlations between GWAS scores calculated on different known AMR gene subsets. We have added these results to the Results section and in greater detail to the Supplemental Analysis.

Results: These results were also used to assess the generalizability of the GWAS score by randomly hiding half of known AMR genes and computing correlations between GWAS scores derived from visible vs. hidden AMR genes. Results across 100 iterations of randomly hiding AMR genes suggests that the GWAS score generalizes well to hidden AMR genes, with Spearman correlation between GWAS scores from visible vs. hidden AMR genes exceeding 0.4 in 6/10 species-drug cases and 0.6 in 4/10 cases (Figure S9, Supplemental Analysis).

Supplemental Analysis: To assess whether a model’s GWAS score calculated from a fixed set of known AMR genes is representative of its capacity to recover other “unseen” AMR genes, the following experiment was conducted. For a given species-drug case, 1) half of all known AMR genes were randomly hidden, 2) GWAS scores were computed using either visible or hidden AMR genes across all 256 tested hyperparameter combinations, and 3) the Spearman correlation was computed between the two GWAS scores as a measure of generalizability. This experiment was conducted for 100 random selections of hidden AMR genes for each of the 10 test species-drug cases (Dataset S5). Overall, the results suggest that the GWAS score often generalizes well to hidden AMR genes, with a median Spearman correlation between GWAS scores from visible vs. hidden AMR genes exceeding 0.4 in 6/10 species-drug cases and 0.6 in 4/10 cases (Figure S9a). The extent of generalizability was dependent on the initial level of GWAS score variation (computed as the standard deviation of GWAS scores across all hyperparameter combinations with all AMR genes visible) (Figure S9b). Low initial

variation in GWAS scores may result in a less robust ranking of models by GWAS score, and consequently weaker correlations between GWAS scores calculated using different subsets of known AMR genes. Conversely, cases of high initial variation in GWAS score, i.e. those where HP optimization can meaningfully impact model performance, are also more likely to have GWAS scores that are representative of the model's performance at recovering yet unknown AMR genes.

Figure S9: Generalizability of the GWAS score. For each species-drug case, half of all known AMR genes were hidden randomly and GWAS scores were computed using either visible or hidden AMR genes for models trained using each possible hyperparameter combination. The distribution of Spearman correlations between the two GWAS scores is shown in (a) for 100 random selections of hidden AMR genes. b) Standard deviation of GWAS scores per species-drug case. c) Distribution of GWAS score Spearman correlations by species-drug case.

- Lines 166 Results and lines 542-543 Methods. How were the 10 “representative” species-drug cases chosen (lines 166 and 542-543)? What does “representative” means? Regarding the 127 species-drug cases, how are these 10 cases connected to them (how well do they represent the 127 cases?)?

Thank you for highlighting this missing detail. The original 127 cases were filtered down to those with substantial data (at least 1000 SIR datapoints and 50 known AMR genes), from which cases were randomly selected while balancing representation of the major drug classes (beta-lactam, aminoglycoside, quinolone, other) and the two remaining phylogenetic classes (Gammaproteobacteria and Bacilli). We have added these details to the Methods. As these cases are not representative of all initial 127 cases (i.e. *Campylobacter* cases are excluded due to low genome count), we have updated the text to refer to them as simply “test cases” rather than “representative cases”, and have also qualified any results that depend directly on analyses of these test cases.

- In Figure S6B and on lines (170-172) the authors claim that a more aggressive feature subsampling increases the number of known AMR genes (GWAS score). However, the authors seem to not provide a statistical test in support of this claim.

Thank you for highlighting this missing test. We have confirmed these results with Mann-Whitney U-tests which have yielded $p < 10^{-15}$ testing for higher GWAS scores using feature sampling at 25% over 50%, 50% over 75%, and 75% over 100%. We similarly confirm that feature subsampling only has a statistically significant impact on model MCC when reduced from 50% to 25%. These results have been added to the Supplemental Analysis, where much of the hyperparameter discussion has also moved.

Supplemental Analysis: Applying Mann-Whitney U-tests to test for differences in GWAS scores from using feature fraction 25% over 50%, 50% over 75%, and 75% over 100% yielded p-values of 3.0×10^{-17} , 6.0×10^{-38} , and 2.6×10^{-29} , respectively. Analogous tests for differences in test set MCC yielded 1.5×10^{-7} , 0.03, and 0.27, respectively, of which only the test between 25% and 50% is significant at FWER < 0.05 (Bonferroni correction, 6 tests).

- When reducing the number of samples for the ensemble, does each base classifier have the same distribution of classes (i.e. resistant and susceptible) with respect to the full set of samples? Or the selection of samples is done randomly, hence no attention is paid to the class distribution? In this case, how does this affect the performance of the ensemble as a whole?

The individual classifiers of the ensembles resample at random without balancing for phenotype or sample phylogenetic class (i.e. MLST subtype). While we did not directly test whether balanced resampling would improve model performance, analysis of the 10 test cases (previously referred to as “representative cases”) suggests that potential

improvements within the scope of our data and analysis. The MCCs (from 5-fold CV) and GWAS scores of ensembles without resampling (using 100% of samples per base classifier) are similar to those with limited resampling (using 75% or 50%), and performance only noticeably drops with more aggressive resampling (25%), which may be due to both the reduced total available samples for training as well as further imbalancing the training set with respect to phenotype (Figure S7, previously Figure S5).

- The authors are emphasizing the use of an ensemble approach; however, they do not demonstrate what the performance of a single SVM compared to the ensemble is like. We conducted an analysis on the performance of single SVMs in our previous work examining 16 species-drug cases (Figure S3 in our previous work³). Individual SVMs from bootstrapping ensembles had out-of-bag MCCs distributed around the test set MCC (from 5-fold CV) of the ensemble as a whole, though with individually poorer recovery of known AMR genes.

- Lines 170-172 Results and lines 542-545 Methods. It is unclear how many known AMR features were present in each one of the base classifiers within the ensemble when the number of features in the ensemble was lower than 100% (range 25%, 50%, 75% and 100%). Did the number of known AMR features affect the performance of the base classifiers? For example, in Figure S5, the models for *S. aureus* cefoxitin achieved a nearly perfect MCC, with almost no AMR features (GWAS score close to 0). Whereas, the models for *S. aureus* ciprofloxacin and erythromycin achieved a lower MCC with much more known AMR features present (GWAS score >4). The same thing happens with *S. enterica*. Shouldn't we expect the opposite (a higher GWAS score leading to higher performance)?

Thank you for highlighting this interesting observation. While we do not have the original features provided to each base classifier for that hyperparameter sweep to evaluate this question directly, we provide two analyses that examine the relationship between known AMR genes available and model performance at the ensemble level. In the response to an earlier comment regarding feature subsampling and GWAS score, we also tested for significant changes in model MCC between reductions in subsampling fractions from 100% to 75%, 75% to 50%, and 50% to 25%, and find that statistically significant reduction in model MCC is only observed between 50% and 25%.

Supplemental Analysis: Applying Mann-Whitney U-tests to test for differences in GWAS scores from using feature fraction 25% over 50%, 50% over 75%, and 75% over 100% yielded p-values of 3.0×10^{-17} , 6.0×10^{-38} , and 2.6×10^{-29} , respectively. Analogous tests for differences in test set MCC yielded 1.5×10^{-7} , 0.03, and 0.27, respectively, of which only the test between 25% and 50% is significant at FWER < 0.05 (Bonferroni correction, 6 tests).

We have also conducted a new analysis between whether the total number of known AMR features impacts the final performance of the ensemble across the 127 species-drug cases in this study (Figure S10a, Table S8, displayed in a previous comment). While GWAS scores were higher with more known AMR features as expected (Spearman correlation 0.294), there was surprisingly a weak negative correlation between the number of known AMR genes and prediction accuracy (test set MCC, Spearman correlation -0.215). We believe these results are in line with previous studies reporting highly accurate AMR models that did not depend on known AMR genes, and highlight the importance of recognizing phenotype prediction accuracy and recovery of true AMR genes as distinct objectives.

Supplemental Analysis: Finally, we note the weak but negative correlation between the number of known AMR genes and model MCC (Spearman correlation = -0.215). This result is consistent with previous ML studies of AMR suggesting that models that accurately predict AMR phenotype often do not rely on known AMR genes and highlights the importance of recognizing phenotype prediction accuracy and recovery of true AMR genes as distinct objectives.

- Lines 172-174 Results. It is unclear how the subset of HP combinations was chosen. The authors seem to select them by using thresholds of the MCC and GWAS score. In my opinion, a statistical test to select the HP combinations that perform statistically better over the 5 cross-validations should be done.

Thank you for this suggestion for a more formal HP selection routine. We hope to implement this approach in future iterations of this work to more precisely select subsets of better performing HP combinations to ultimately improve final model performance and/or overall runtime of this workflow.

For the purposes of this study, we have revisited this analysis to test whether our chosen subset of HP combinations yields statistically better performance than the HP combinations excluded (Table S7). For each of the 10 species-drug cases for which a full HP combination sweep was tested, the test set MCCs and GWAS scores across all folds were combined across all HP combinations included in our subset, and separately for all combinations excluded from our subset. Applying Mann-Whitney U-tests, the performance on our subset yielded statistically different performance from excluded combinations in 2/10 cases for MCC and 5/10 for GWAS score (FWER < 0.05, Bonferroni correction). In 2/2 significant MCC cases and 3/5 significant GWAS score cases, the subset had an equal or higher median score than the excluded combinations. These results are in line with our original main goal of reducing the number of HP combinations to test to reduce overall runtime, at the expense of some optimal solutions in a few rare cases (i.e. the two cases where the excluded combinations had significantly higher GWAS scores). We have added these results to the Supplemental Analysis.

Supplemental Analysis: The impact of reducing the number of tested HP combinations was examined as follows. For each of the 10 test species-drug cases, the test set MCCs and GWAS scores across all folds (from 5-fold cross validation) were combined across all models resulting from tested HP combinations, then split into those associated with included HP combinations and those associated with excluded combinations. Mann-Whitney U-tests were applied to determine whether the MCCs or GWAS scores differed significantly between models with included or excluded HP combinations (Table S7). Significant differences were observed in 2/10 cases for MCCs and 5/10 cases for GWAS scores (FWER < 0.05, Bonferroni correction, 20 tests). In 2/2 significant MCC cases and 3/5 significant GWAS score cases, the subset had an equal or higher median score than the excluded combinations, suggesting that the reduction in the set of tested HP combinations only rarely reduces the maximum performance attainable.

Table S7: Statistically significant differences in performance between selected and excluded hyperparameter combinations for 10 species-drug cases. For each case, a Mann-Whitney U-test was conducted to determine if the trained SVM model’s test set MCCs or GWAS scores, across all hyperparameter combinations (HPCs) and folds from 5-fold CV, differed significantly for selected HPCs than for excluded HPCs. See Table S6 for HPC sets. Starred cases are significant to FWER < 0.05, Bonferroni correction (20 tests).

AMR Case	Test set MCC			GWAS Score		
	Included Median	Excluded Median	MW test p-value	Included Median	Excluded Median	MW test p-value
A. baumannii , amikacin	0.83782	0.83815	0.23016	4.04567	4.07742	0.04913
A. baumannii , ceftazidime	0.76253	0.75629	0.00138*	0.9075	0.59558	<0.00001*
E. faecium , ampicillin	0.95698	0.95698	0.12555	1.04808	1.07578	0.1873
E. coli , gentamicin	0.86522	0.86143	<0.00001*	4.68256	5.1806	<0.00001*
K. pneumoniae , ciprofloxacin	0.81835	0.81573	0.41189	0.88562	0.87238	0.34162
S. enterica , ceftriaxone	0.9751	0.9751	0.04874	2.74439	3.12598	<0.00001*
S. enterica , chloramphenicol	0.93276	0.93276	0.12058	4.24349	4.28149	0.77218
S. aureus , ceftazidime	1.00000	1.00000	0.94434	0.00000	0.00000	<0.00001*
S. aureus , ciprofloxacin	0.96917	0.96917	0.60248	4.94068	4.67576	0.00018*
S. aureus , erythromycin	0.96318	0.9575	0.04671	6.23253	6.23332	0.37441

- Lines 175-176 Results. The authors do not explain why 10 known AMR genes are sufficient to include the datasets with a minimum of 10 known AMR genes in the analysis. Also, the authors use a minority phenotype of > 5%. This could be an issue for the SVMs as explained below (see imbalance data).

Thank you for this observation. Both of these thresholds were chosen to be unrestrictive to maximize the number of species-drug test cases against which to evaluate our workflow, and a dedicated optimization of these thresholds would be a valuable addition in a future iteration of this work. However, as we find only a weak correlation between model performance and minority phenotype fraction (Figure S10a, Table S8, Spearman correlation 0.005 with test MCC, 0.250 with AMR genes recovered) and between model performance and number of known AMR genes (Figure S10a, Spearman correlation -0.215 with test MCC, 0.294 with AMR genes recovered), we believe it is unlikely that

additional tuning of these thresholds will significantly impact the final list of AMR gene candidates identified. We have added these new results regarding the number of known AMR genes to the Supplemental Analysis.

- Lines 192-197 Results. The authors mentioned that similar features were obtained when using the ensemble and Fisher's exact test. Have the authors tested the performance of the ensemble when using the features obtained with Fisher's exact test as input and compared it to the performance obtained with the pipeline they are proposing? How do the two pipelines compare?

Thank you for this interesting suggestion on integrating these two methods. Given the similarity of this adjustment to that of altering the LOR threshold for filtering input features, we examined both suggestions together in the response to the later comment regarding the LOR threshold.

- Lines 434 -onwards the authors state that:" Briefly, protein sequences were clustered using CD-HIT v4.6 with minimum identity 80% and minimum alignment length 80%²⁴." And at line Line 494-onwards the authors state that:" For machine learning purposes, all features associated with a gene cluster containing a sequence linked to resistance for a drug by RGI or PATRIC text annotation were treated as known AMR features". The genes within the same cluster can vary up to 20% and such variation can include deletions, insertions, etc., which may cause a reduced/inactivated gene with respect to the linked resistance. However, all the clustered sequences are still treated equally as known functional AMR genes. Can the authors prove that such variability is not associated with reduced resistance phenotype and that those genes are truly underlying the targeted AMR profile? Can the authors justify why an 80% identity threshold is a good choice with respect to other homology values for clustering AMR features to be used for machine learning?

The similarity threshold is indeed a parameter with significant implications throughout the workflow and would benefit from a detailed examination with respect to AMR. While ideal, it was not feasible for us to comprehensively validate every sequence variant we labeled as a known AMR feature for experimental evidence confirming its contribution to resistance. Consequently, we assumed that sequences sufficiently similar to AMR genes confirmed by RGI (i.e. within similarity thresholds previously examined in pangenome studies for ortholog assignment) were likely to also be AMR genes. The 80% used in this study is based on the threshold from our previous pangenome study on the same 12 species ⁵, and falls in the range of values commonly used for pangenome analysis, ranging from 70% for an *E. coli* pangenome⁹ to as high as 95% in the Roary pangenome pipeline¹⁰.

To examine this assumption, we have added a new analysis of all instances of our models associating a *gyrA* allele with fluoroquinolone resistance (30 alleles across 7

species), which had originally inherited the “known AMR” label from wildtype *gyrA* alleles annotated by RGI. All alleles positively associated with resistance contained at least one known resistance-conferring mutation, while those negatively associated contained no such mutations (see Supplemental Analysis and Dataset S5), and all alleles were within five mutations of the wildtype allele. Based on this result, we believe our AMR gene annotation approach is reliable (at least for AMR genes recovered by SVM) and is unlikely to overestimate the capabilities of the SVM at recovering known AMR genes.

Results: Finally, a detailed analysis of the 30 *gyrA* alleles identified by SVM as associated with fluoroquinolone resistance finds all such variants to be consistent with the current literature on *gyrA*-mediated resistance; all positively-associated alleles carried at least one known resistance-conferring substitution, while all negatively-associated alleles carried no such mutations (Supplemental Analysis, Dataset S5).

Supplemental Analysis: All *gyrA* alleles among the top 50 features from models related to fluoroquinolone resistance were identified, totaling 30 alleles across 7 species. Mutations were called relative to the wildtype allele of the corresponding species, defined as the most commonly observed *gyrA* allele among genomes susceptible to at least one fluoroquinolone. Resulting mutations and GenBank accession IDs for wildtype alleles are available in Dataset S5. All recovered *gyrA* alleles with positive feature weight for resistance had at least one known resistance-conferring mutation, covering the following substitutions: S81L in *A. baumannii*, T86I in *C. coli* and *C. jejuni*, S83L and D87N in *E. coli*, T83I in *P. aeruginosa*, and S83Y and D87* in *S. enterica*. All recovered *gyrA* alleles with negative feature weight for resistance had no such known resistance-conferring mutations. These results suggest that the *gyrA* alleles recovered by the SVM ensemble approach are consistent with current understanding of the *gyrA* mutational landscape with respect to fluoroquinolone resistance.

- In line 527 Methods the authors state: “1) features present or missing in less than 3 genomes were removed”, can the authors show the distribution of the features across the resistant/susceptible genomes considered, and how such distribution look after the classification? If a feature is associated with a specific resistance, how many genomes with that specific resistant phenotype (CLSI, etc.) out of the total contain that feature? and likewise how many susceptible labelled genomes (CLSI, etc.) for that specific antibiotic contain/do not contain that feature?

To examine the distribution of resistant/susceptible genomes associated with features we labeled as “known AMR”, we have computed the fraction of all genomes with each feature that are resistant to the relevant drug (“resistance fractions”), and have visualized these distributions for features present/missing in less than 3 genomes, and for all others (Figure S14). These resistance fraction distributions tended to concentrate near 0 and 1 with relatively fewer genes closer to 0.5, suggesting that the features

labeled as known AMR genes are more likely than not related to resistance and are correctly labeled (features with resistant fractions closer to 1 are more likely to be resistance determinants, while those with fractions closer to 0 are more likely markers for susceptibility such as a known fluoroquinolone-susceptible *gyrA* allele). We have added these results to the Supplemental Analysis.

Finally, for our proposed AMR gene candidates, each candidate is strongly enriched for resistant genomes across multiple drugs (with log2 odds ratios typically exceeding 3 for at least two drugs from the same class), in line with our selection criteria which examined statistical evidence across related drugs. LOR values for the final candidates are available in Dataset S7, and for all SVM model top features in Dataset S5.

Supplemental Analysis: For each known AMR feature across the 127 species-drug cases examined using the SVM workflow, the set of genomes carrying the feature was identified and the fraction of such genomes that are resistant to the corresponding drug was computed. The distribution of these resistance fractions was computed for each species-drug case individually and in aggregate, combining all known AMR features across all cases (Figure S14). The distributions generally concentrated around resistance fractions of 0 or 1, which was quantified by computed the mean of resistance fractions for values either less than or greater than 0.5, for each species-drug case (Figure S14b-c). These results suggest that features labeled as “known AMR” are more likely than not related to resistance and are correctly labeled, i.e. features with resistance fractions closer to 1 are more likely to be resistance determinants, while those with fractions closer to 0 are more likely to be markers for susceptibility such as known fluoroquinolone-susceptible *gyrA* alleles.

Figure S14: Associations between known AMR features and resistance. For each known AMR feature in a given species-drug case, the fraction of genomes with the feature that are resistant to the drug was computed. The distribution of these resistance fractions combined across 127 species-drug cases is shown in (a), with black dots representing rare features (found in no more than 2 genomes) and blue bars representing all other known AMR features. The tendency for these distributions to concentrate near 0 and 1 was quantified by computed means for values b) less than or c) greater than 0.5. Individual distributions for 10 species-drug cases are shown in (d).

- Line 528..” remaining features were sorted by log odds ratio (LOR) for resistant genomes, and features with the 25,000 highest and 25,000 lowest LORs were retained.” Did the authors test what happens when changing the threshold of the odds ratio? How does the performance of the ensemble change when selecting a lower threshold (less than 25.000)? And with this lower threshold are the features proportionally more connected to known AMR genes?

Thank you for these suggestions regarding adjustments to how input features to the workflow are determined (altering LOR threshold, or using Fisher’s exact test p-values instead as mentioned in a previous comment). As it is computationally infeasible for us to repeat the analysis on all species-drug cases for variations on input filters, we have conducted such an analysis for 10 test cases. Specifically, we trained models across the original set of 256 HP combinations, provided with either the top 5000, 10000, 20000, or 50000 features when sorted by either LOR or Fisher’s exact p-value, (8 input feature sets tested). Each input feature set was evaluated by computing the median and maximum test set MCCs (5-fold CV) and GWAS scores across all HP combinations (Figure S15).

Broadly, we find that in most of the tested species-drug cases, the maximum and median MCC and GWAS scores are robust to the number of features provided, with a few exceptions under Fisher’s exact test filters demonstrating an increasing trend (more features = better predictive performance and/or AMR gene recovery). We believe this suggests that most, but not all, AMR models will not benefit from accessing features with individually low statistical signals, and that future iterations of our workflow could implement stricter feature filtering to improve runtime with relatively little cost to model performance. When comparing equal size feature sets generated from filtering by either LOR or Fisher’s exact test, the models derived from the LOR filter consistently performed better. This may be due to the LOR’s emphasis on effect size relative to Fisher’s exact test p-values.

Supplemental Analysis: Variations to the approach of pre-filtering features by log odds ratio (LOR) were evaluated for their impact on downstream model performance. The existing filter of taking the top 50,000 features by LOR (specifically, the features with the 25,000 highest and lowest LORs) was compared to analogous filters taking the top 20,000, 10,000, or 5,000 features by LOR. These were also compared to filters taking the top 50,000, 20,000, 10,000 or 5,000 features by Fisher’s exact test p-value when testing for enrichment in resistant genomes, for a total of eight possible feature filters. After applying each filter, SVM ensembles were trained to predict AMR phenotype across the 10 test species-drug cases and 256 HP combinations described in the previous section. The maximum and median test set MCC (from 5-fold cross validation) and GWAS score was computed for each species-drug case and under each filter (Figure S15).

In most of the tested species-drug cases, the maximum and median MCC and GWAS scores are robust to the number of features provided, with a few exceptions under Fisher's exact test filters: Model MCCs for the *E. coli*-gentamicin and *S. aureus*-erythromycin cases and GWAS scores for the *A. baumannii*-amikacin, *S. aureus*-ciprofloxacin, and *S. aureus*-erythromycin cases increase with the number of features. When comparing equal size feature sets generated from filtering by either LOR or Fisher's exact test p-value, models derived from the LOR filter consistently performed better. These results suggest that 1) future applications of this ML workflow may be able to apply stricter preliminary feature filters to reduce the computational resources required for training with relatively little impact on performance, and 2) the LOR appears to more effectively identify features required for better performing models than the Fisher's exact test p-value, possibly due to its emphasis on effect size over significance.

Figure S15: Impact of varying the preliminary feature filter on downstream model performance. Each row shows the effect of the feature filter on a model performance metric, either the maximum or median test set MCC (from 5-fold CV) or GWAS score across 256 hyperparameter combinations. The first column shows model performance when limited to the top 5,000, 10,000, 20,000, or 50,000 features after sorting by log odds ratio (LOR) for resistance, and the second column shows model performance under analogous filters sorting instead by Fisher's exact test p-value for resistance. The third column compares performance between LOR and Fisher's exact test filters for identical feature count limits. Each color represents results for a specific species-drug case.

- Lines 530-531 Methods. It is unclear how the selection of features and samples was done. Are the authors using replacement for both the selection of features and the selection of samples? On lines 530-531 the authors mention that they are using the BaggingEnsemble function from sklearn, however they do not provide the version of sklearn that they used. On the newest version 1.2.1 there is no BaggingEnsemble function only the BaggingClassifier function (<https://scikit-learn.org/stable/modules/generated/sklearn.ensemble.BaggingClassifier.html>). On the BaggingClassifier function only the samples are selected with replacement, the features are not (if the default values are used).

Thank you for catching this error, this was likely a typo and we have updated the Methods to indicate that we used BaggingClassifier from scikit-learn v1.0.1. We have also confirmed and clarified in the Methods that samples were selected with replacement and features without replacement as is default for BaggingClassifier.

- Lines 530-532. The authors as done in previous papers used a linear SVM. However, here they do not provide a reason for using a linear SVM. Some of the known disadvantages of using a linear SVM are 1. they do not perform well when the classes are overlapping (since the authors are using the intermediates, which could be a concern because the intermediates could be more closely related to susceptible samples) and 2. SVMs may present suboptimal results when dealing with imbalanced datasets (Table S4 indicates that 9 out of the 10 cases have imbalanced data).

Thank you for these detailed comments on known drawbacks of linear SVMs. While we have had some success in improving the performance of SVMs on imbalanced data through weighting the loss function with respect to class frequency (described in the Methods and employed both here and in our previous works), we had not considered the specific issues that overlapping classes could cause for SVMs. The choice of linear SVM was largely based in developing incremental improvements and implementing larger scale demonstrations of our previous ML-AMR workflows that began with SVMs. We have also examined the impact of intermediate genomes on model performance in a response to an earlier comment.

We also agree that other model architectures will likely provide better performance than SVMs in light of the issues you mention, and include in the Discussion that evaluating other models within a similar framework would be a valuable extension of this work. Nonetheless, we believe that the promising results shown here (which are now further supported by a favorable comparison against Pyseer) demonstrate the value and robustness of linear SVMs in conducting GWAS for AMR.

- Regarding the experimental validation of cycA and frdD, can the authors justify why these two were selected?

Thank you for indicating the lack of detail in this selection. We have added a new section

to the Supplemental Analysis (“Selection of AMR gene candidates for experimental validation”) that describes additional decisions not previously mentioned in the Methods as well as intermediate results throughout the process of translating the top features from our ML models, to the 142 proposed candidates, and to the final choice of *frdD* and *cycA* for validation. In brief, the choice was based on an examination of functional annotations and cross-drug evidence not included in the ML training process, as well as our own limitations in the types of features we were best equipped to test experimentally.

In more detail, the initial reduction from the top model features to the 142 candidates follows Figure 4a and is detailed in Dataset S7. From top model features, those associated with known AMR genes or observed very rarely were removed first, and remaining features were scored based on 1) the number of drugs from the same drug class with which they were significantly associated with resistance, and 2) the number of drugs for which these exist resistant genomes without any known AMR genes that could be explained by the feature. The top 10 features by this score for each (species, drug class) pairs yielded the 142 candidates.

These were further filtered to exclude features that were poorly annotated, related to mobile elements, or associated with AMR for unrelated drugs to yield a smaller set of 43 “well-characterized” candidates (also detailed in Dataset S7). Just six of these referred to specific sequence variants in *E. coli* for which we were best equipped to validate experimentally. We ultimately decided to focus on coding variants to take advantage of the Keio knockout collection which left us with four options, from which we chose *frdD* and *cycA* due to their better functional and metabolic characterization compared to *sugE* and *yjfN*.

Response to Reviewers - References

1. Hawkey, J. *et al.* Global phylogenomics of multidrug-resistant *Salmonella enterica* serotype Kentucky ST198. *Microb Genom* **5**, (2019).
2. Enoma, D. O., Bishung, J., Abiodun, T., Ogunlana, O. & Osamor, V. C. Machine learning approaches to genome-wide association studies. *J. King Saud Univ. Sci.* **34**, 101847 (2022).
3. Hyun, J. C., Kavvas, E. S., Monk, J. M. & Palsson, B. O. Machine learning with random subspace ensembles identifies antimicrobial resistance determinants from pan-genomes of three pathogens. *PLoS Comput. Biol.* **16**, e1007608 (2020).
4. Parrello, B. *et al.* A machine learning-based service for estimating quality of genomes using PATRIC. *BMC Bioinformatics* **20**, 486 (2019).
5. Hyun, J. C., Monk, J. M. & Palsson, B. O. Comparative pangenomics: analysis of 12 microbial pathogen pangenomes reveals conserved global structures of genetic and functional diversity. *BMC Genomics* **23**, 7 (2022).
6. WHO publishes list of bacteria for which new antibiotics are urgently needed. <https://www.who.int/en/news-room/detail/27-02-2017-who-publishes-list-of-bacteria-for-which-new-antibiotics-are-urgently-needed>.
7. Huang, Y. *et al.* Urinary excretion of D-serine in human: comparison of different ages and species. *Biol. Pharm. Bull.* **21**, 156–162 (1998).
8. Connolly, J. P. R. *et al.* A Highly Conserved Bacterial D-Serine Uptake System Links Host Metabolism and Virulence. *PLoS Pathog.* **12**, e1005359 (2016).
9. Monk, J. M. *et al.* Genome-scale metabolic reconstructions of multiple *Escherichia coli* strains highlight strain-specific adaptations to nutritional environments. *Proc. Natl. Acad. Sci. U. S. A.* **110**, 20338–20343 (2013).
10. Page, A. J. *et al.* Roary: rapid large-scale prokaryote pan genome analysis. *Bioinformatics* **31**, 3691–3693 (2015).

REVIEWERS' COMMENTS

Reviewer #2 (Remarks to the Author):

My concerns have been addressed.

Reviewer #3 (Remarks to the Author):

The authors have comprehensively addressed the comments raised in my review. I have no further comments.